# RETRIEVAL-BASED GENERALIZED CROWD COUNTING

## ABSTRACT

Existing crowd-counting methods rely on the manual localization of each person in the image. While recent efforts have attempted to circumvent the annotation burden through vision-language models or crowd image generation, these approaches rely on pseudo-labels to perform crowd-counting. Simulated datasets provide an alternative to the annotation cost associated with real datasets. However, the use of large-scale simulated data often results in a distribution gap between real and simulated domains. To address the latter, we introduce knowledge retrieval inspired by knowledge-enhanced models in natural language processing. With knowledge retrieval, we extract simulated crowd images and their text descriptions to augment the image embeddings of real crowd images to improve generalized crowd-counting. Knowledge retrieval allows one to use a vast amount of non-parameterized knowledge during testing, enhancing a model's inference capability. Our work is the first to actively incorporate text information to regress the crowd count in any supervised manner. Moreover, to address the domain gap, we propose a pre-training and retrieval mechanism that uses unlabeled real crowd images along with simulated data. We report state-of-the-art results for zero-shot counting on five public datasets, surpassing existing multi-model crowd-counting methods. The code will be made publicly available after the review process.

## 1 INTRODUCTION

Crowd-counting has garnered significant interest owing to its extensive applications in safety and population management (Sindagi & Patel, 2018; Kang et al., 2018). Accurately estimating counts becomes particularly challenging, especially in densely populated areas.

Most prominent crowd-counting methods either estimate a density map (Sindagi & Patel, 2017; Ranasinghe et al., 2024; Han et al., 2023) or localize head positions (Song et al., 2021; Liang et al., 2022b) to estimate the count. However, these methods require point-level annotations for human heads, which is an expensive and laborious process. Recently, to relieve the cost of annotation, the field has been moving towards using vision-language models and synthetic images. An illustrative example of this trend is observed in the introduction of the CrowdCLIP (Liang et al., 2023) model, which integrates the CLIP (Radford et al., 2021) architecture for crowd-counting showcasing a contemporary approach in merging vision and language models for this specific task. While CrowdCLIP is positioned as an unsupervised model requiring no explicit count labels, evaluating the test set involves determining the optimal count label structure for performance assessment. In contrast, the AFreeCA (D'Alessandro et al., 2024) model proposes a fully supervised crowd-counting strategy by synthesizing crowd images using stable diffusion and multi-modal supervision. However, a notable challenge arises in AFreeCA, where the actual crowd count in the synthesized images diverges from the count provided as the text condition to the model, introducing inherent noise into the pipeline.

However, CrowdCLIP and AFreeCA demonstrate the transferability and generalizability of incorporating text knowledge and a vast amount of data to annotator-free crowd-counting. Consequently, we can address the annotation cost involved in crowd-counting by training a model with simulated data to perform zero-shot crowd-counting on real images. The benefits of using simulated data are two-fold: 1. we can create point annotations without any human labor. 2. We can create a huge amount of data for the model to train. Naturally, models need more capacity to parameterize a large corpus of data, as evidenced by large language models. However, recently developed retrieval augmented generation (Lewis et al., 2020) for natural language processing demonstrated the advantage of using non-parametric knowledge (external information) for more updated, reliable response generation. Following this, RA-CLIP (Xie et al., 2023) illustrated the advantage of using a reference database for zero-shot performance with vision-language models for classification. However, the

benefits of retrieval-based models have not been studied for regression-based downstream tasks, let alone for crowd-counting.

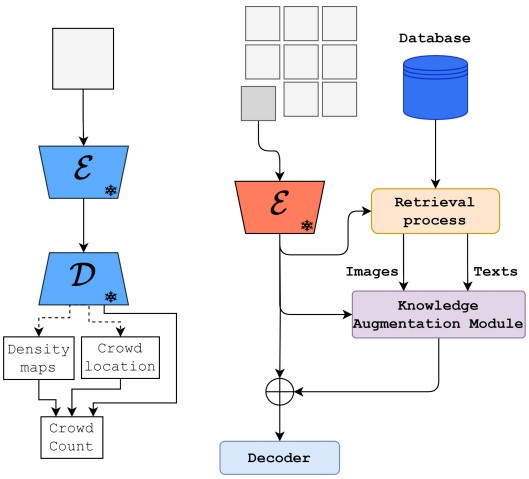

Figure 1: (a) The fully parameterized supervised methods require point annotations for real crowd images, which need heavy manual labor to label a large-scale dataset. (b) Vision-language contrastive training to learn counting labels. (c) The proposed vision-language enhanced training for generalized counting without labeled real crowd data.

In this paper, we propose ReGe-Count, which combines vision-language retrieval of simulated crowd data to estimate the crowd count of real crowd images under the zero-shot scenario. With ReGe-Count, we demonstrate the benefit of using multi-modal context for real crowd images to perform regression-based crowd-counting with weak supervision. Specifically, first, we train an image encoder to parameterize visual understanding of the unlabeled real and simulated crowd images under the self-supervised objective of ranking. Second, we train a knowledge augmentation module to extract information from image-text pairs from the simulated dataset for a given query image as displayed in the right-most figure of figure 1. This retrieved image-text information is combined with the query embeddings to learn the mapping between crowd semantics and crowd count under weak supervision. Furthermore, unlike Crowd-CLIP, we don't need a progressive refinement strategy to remove ambiguous crowd patches, which jettison the necessity to pass the image crops through the image encoder multiple times. Besides that, CrowdCLIP does not utilize language understanding to produce the crowd count; instead, the CrowdCLIP pipeline classifies the image patches into different classes at different stages without the need to understand the class label information.

Comprehensive experiments carried out across five datasets in diverse scenarios underscore the efficacy of our ReGe-Count. Notably, our approach outperforms the current state-of-the-art annotator-free methods on public crowd-counting datasets, as measured by the MAE metric. Our major contributions in this paper can be summarized as follows: 1) We propose knowledge retrieval for crowd estimation with regression. To the best of our knowledge, this is the first work to utilize external sources at testing to enhance crowd-counting. 2) We introduce combining vision-language information for weakly-supervised crowd-counting. This is one of the first works to utilize and infuse language understanding into crowd-counting. 3) We successfully demonstrate using simulated labeled crowd images for generalized crowd-counting of real crowd images, surpassing the zero-shot performance of other vision-language crowd-counting methods.

## 2 RELATED WORKS

**Annotator-free crowd-counting.** Existing crowd-counting methods use real-world images with manually annotated ground truth, a labor-intensive and costly process. To mitigate the dependence on these human annotations, recent research has explored annotator-free approaches to crowd-counting. For example, CSS-CCNN utilizes self-supervised learning by pretraining the image encoder with a rotation prediction task before fine-tuning the encoder and a density decoder using Sinkhorn matching, completely bypassing ground truth annotations. Similarly, CrowdCLIP leverages the CLIP architecture to train an image encoder for crowd interval prediction by contrasting image features with count interval labels. In contrast to these methods, AFreeCA performs fully supervised crowd-counting by training its network on synthetic images generated using stable diffusion, enabling the model to learn crowd counts directly from artificially generated data.

**Real and simulated crowd images.** Text descriptions about images establish the multi-modal relationship among image-text pairs. However, these text descriptions generally include information about objects present in the image and the context of the image. But specific information like the

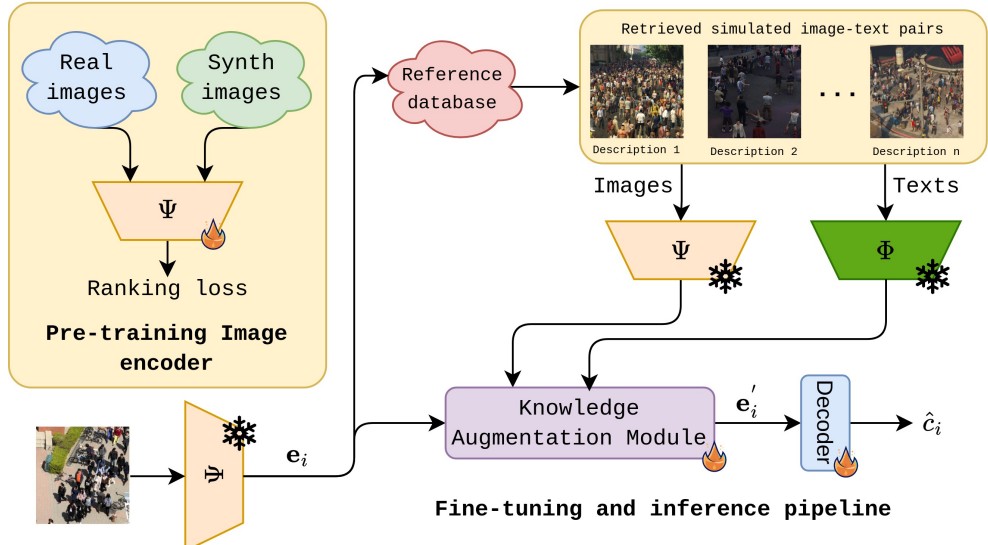

Figure 2: Overall pipeline of ReGe-Count. First, the image encoder ($\Psi$) is pre-trained using both real and simulated images using the ranking loss. Next, the Knowledge Augmentation Module (KAM) and the count decoder are trained during the fine-tuning stage. During the fine-tuning stage $\Psi$ and the pre-trained text encoder ($\Phi$) are frozen.

actual crowd count is required for crowd-counting. Hence, one must manually annotate the images to get the crowd count, which is a tedious and time-consuming task (D'Alessandro et al., 2024). In contrast, using simulated data (Wang et al., 2019) eliminates the necessity of labor for annotation and caption generation. This is because the context, conditions, and crowd locations are readily available when preparing the simulated images, unlike real crowd images.

**Non-parametric knowledge retrieval.** Recently, knowledge-enhanced models have been gaining traction in the vision domain after its success with large-language models (Lewis et al., 2020). First, Hu et al. (2023) improves the performance of visual question answering by storing image-text pairs in an external database and training a network to extract relevant knowledge to enhance model responses. Then, Xie et al. (2023) improves the zero and few-shot performance of the CLIP model by augmenting the input image embeddings with image-text pair information from an external database. In addition, Chen et al. (2024) and Liu et al. (2023) utilize the knowledge-enhanced models to improve classification performance with diffusion models and customized visual models. However, the above methods cater to classification for a given image. In our work, we use external knowledge retrieval to improve the performance of crowd-counting in a weakly supervised learning manner.

## 3 PROPOSED METHOD

The overall idea of our proposed framework is to retrieve image-text information from an external database for a given query image to enhance the inference performance for crowd counting as shown in figure 2. We first discuss constructing the external database and the image-text data in section 3.1. Next, we discuss the retrieval process in section 3.2, and the knowledge augmentation in section 3.3.

### 3.1 REFERENCE SET CONSTRUCTION

**Text descriptions for simulated images.** We utilize the crowd locations, weather conditions, and time conditions available for each crowd image in the GCC dataset. For each crowd image, we construct a text description like,

"The image has a [weather condition] weather with [crowd count] people in the [time of day]."

For the weather conditions, we use {clear, cloudy, rainy, foggy} as the labels, and for the time of day, we use {morning, evening, night} as the prompts. These text descriptions are only constructed for the images that will be included in the reference database. We use $80\%$ of images from the GCC training set for the reference database. However, including only simulated data in the reference set introduces a domain gap between simulated and real-world test images.

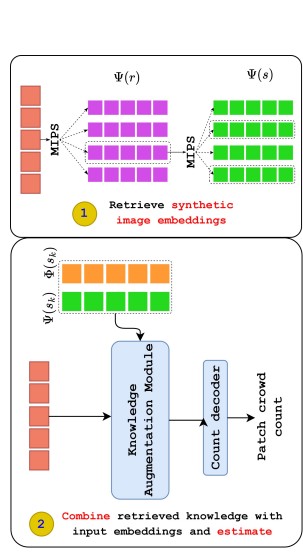 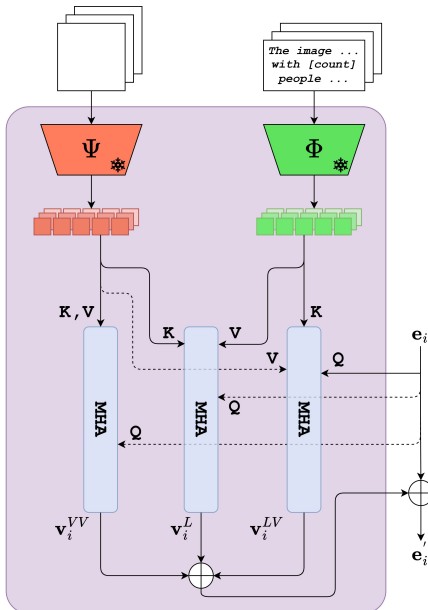

(a) Retrieval and augmentation pipeline  (b) Knowledge augmentation module

Figure 3: (a) In the retrieval and augmentation pipeline, we first do maximum inner product search (MIPS) between query image embeddings ($\mathbf{e_i}$) and real image embeddings ($\Psi(r)$). For the most similar $\Psi(r)$, find the closest simulated embeddings ($\Psi(s)$) using MIPS. These retrieved image-text embeddings ($\Psi(s_k)$ and $\Phi(s_k)$) and $\mathbf{e}_i$ are passed through the augmentation module to get the count. (b) The network flow of the augmentation module to extract non-parameterized knowledge.

Hence, to align the distribution of simulated and real data, we embed the simulated images in the latent space of real images.

**Real crowd image set construction.** For the real crowd image set, we combine the existing publicly available crowd counting datasets except for the dataset of which the performance is evaluated. *e.g.*, suppose we evaluate the performance on ShanghaiTech Part-A, then the real crowd images in the reference set will contain crowd images from ShanghaiTech Part-B, JHU-Crowd++, UCF-QNRF, and NWPU-Crowd. This ensures that the image encoder has not seen any images from the test distribution, unlike CrowdCLIP and AFreeCA.

**Image encoder pre-training.** We pre-train the image encoder using both real and simulated crowd images with the ranking loss (Liu et al., 2018), which does not require any labels. Ranking loss has been used to pre-train the image encoder in recent multi-modal crowd counting works like CrowdCLIP and AFreeCA. To construct the ranking crops for the pre-training of the image encoder, we follow the sampling procedure provided in Liu et al. (2018) and moderate it for the simulated and real images separately. We pass the image embeddings of each crop through a linear layer to map it to a count value. To enforce the ranking, we apply the pairwise ranking hinge loss, which for a single pair is defined as:

$$L_r = \max\left(0, \hat{c}(I_l) - \hat{c}(I_h)\right), \tag{1}$$

to penalize incorrect ranking pairs, where $\hat{c}(I_l)$ is lower than $\hat{c}(I_h)$, and $I_l$ and $I_h$ represent two ranking patches from the image. It should be noted that the $L_r$ loss is proportional to the difference between the estimates when the two estimates don't obey the correct ranking order and help embed the real and simulated images into an ordinal space (Li et al., 2022). The image encoder is trained using the gradient updates given as:

$$\nabla L_r = \begin{cases} 0 & \text{if } \hat{c}(I_l) - \hat{c}(I_h) \leq 0 \\ \nabla\hat{c}(I_l) - \nabla\hat{c}(I_h) & \text{otherwise} \end{cases} \tag{2}$$

with respect to the image encoder parameters. For a given image, we combine the losses of each pair before taking the gradients. There will be $\binom{M}{2}$ pairs, where $M$ is the crops per image.

**Reference vector database.** After training the image encoder, we construct image crops of size $224 \times 224$ from simulated and real crowd images. Next, we collect the image embeddings of these

crops to create a vector database to perform knowledge retrieval under the maximum inner product.

### 3.2 IMAGE-TEXT RETRIEVAL

In knowledge retrieval, we extract $K$ image-text pairs for an input image $I_i$. First, we get the image embeddings $\mathbf{e}_i$ of $I_i$ using the pre-trained image encoder. Then, we perform the maximum inner product search (MIPS) (Yu et al., 2017) with the image embeddings of the real crowd image crops in the reference database. The MIPS is formulated such that, given a query vector ($q \in \mathbb{R}^d$) and a set of data vectors ($\mathcal{V} = \{v_1, v_2, \ldots, v_n\} \subset \mathbb{R}^d$), to find the nearest $k$-vectors ($\mathcal{V}^* = \{v_1^*, v_2^*, \ldots, v_k^*\} \subset \mathcal{V}$) to $q$. $\mathcal{V}^*$ should be found such that:

$$\langle q, v_i^* \rangle \geq \langle q, v_j \rangle, \quad \forall v_j \notin \mathcal{V}^*, \quad \text{and} \quad |\mathcal{V}^*| = k,$$

where $\langle \cdot, \cdot \rangle$ and $| \cdot |$ represent the vector inner product and cardinality. From the search, we find the $K/2$ most similar real image crops. Next, for each real crowd crop, we will find the 2 most similar simulated crowd crops without any repetitions. *i.e.*, if any two real crops share a simulated crowd crop, we assign the simulated crop to the real crop that has the highest inner product with $I_i$ and assign the next most similar simulated crop to the remaining real crop. Once we find $K$ most similar simulated crop vectors, we extract their corresponding image crops $\{\mathbf{r}_k^{V_i}\}_{k=1}^K$ and the text descriptions $\{\mathbf{r}_k^{L_i}\}_{k=1}^K$ from our reference set. In the retrieval process, the reason to extract real crops first is to align the simulated crops retrieved for real crowd images during testing. Since we train the knowledge augmentation module (see section 3.3) and the count decoder with simulated data, if we directly extract the $K$ most similar simulated crops from the reference set, the strong relationship of being from the same domain will not exist during testing with the real images. However, by using real crops as an intermediary, we can alleviate this issue as the connection between the retrieved simulated crops and input image will be stronger during testing since the test image and intermediary reference crops are from the same domain. This intermediary process can be considered as a projection of the input image features onto the real image features.

### 3.3 KNOWLEDGE AUGMENTATION MODULE (KAM)

The overall architecture of the KAM is illustrated in figure 3b. In the KAM, we first extract the image embeddings and the text embeddings for the retrieved image-text pairs ($\{\mathbf{r}_k^{V_i}\}_{k=1}^K$ and $\{\mathbf{r}_k^{L_i}\}_{k=1}^K$) using the pre-trained image encoder ($\Psi$) and a pre-trained text encoder ($\Phi$) as follows,

$$\mathbf{h}_k^{V_i} = \Psi\left(\mathbf{r}_k^{V_i}\right) \text{ and } \mathbf{h}_k^{L_i} = \Phi\left(\mathbf{r}_k^{L_i}\right), \tag{3}$$

where $\mathbf{h}_k^{V_i}$ and $\mathbf{h}_k^{L_i}$ represent the image embeddings and text embeddings of the $k^{\text{th}}$ image-text pair. Since $\Psi$ and $\Phi$ are pre-trained encoders, the embeddings in equation 3 can be pre-computed as these models are frozen during training of the KAM and the count decoder.

Once the reference image embeddings $\{\mathbf{h}_k^{V_i}\}_{k=1}^K$ and the corresponding text embeddings $\{\mathbf{h}_k^{L_i}\}_{k=1}^K$ are extracted, we infuse this external knowledge to the input image embeddings ($\mathbf{e}_i$) using Multi-head Attention (MHA) (Vaswani et al., 2017) in the KAM. First, we take $\mathbf{e}_i$, $\{\mathbf{h}_k^{V_i}\}_{k=1}^K$, and $\{\mathbf{h}_k^{L_i}\}_{k=1}^K$ as the *query*, *key*, and *value*, respectively to produce text-knowledge-infused embeddings ($\mathbf{v}_i^L$) given by,

$$\mathbf{v}_i^L = \text{MHA}(\mathbf{e}_i, \{\mathbf{h}_k^{V_i}\}_{k=1}^K, \{\mathbf{h}_k^{L_i}\}_{k=1}^K). \tag{4}$$

Here, input image embeddings will learn the weight aggregation of $\{\mathbf{h}_k^{L_i}\}_{k=1}^K$ depending on the relationship between $\mathbf{e}_i$ and $\{\mathbf{h}_k^{V_i}\}_{k=1}^K$. Similarly, we also produce image-knowledge-infused embeddings from the KAM. However, unlike $\mathbf{v}_i^L$, here we can use both $\{\mathbf{h}_k^{V_i}\}_{k=1}^K$ and $\{\mathbf{h}_k^{L_i}\}_{k=1}^K$ as *key* while $\mathbf{e}_i$ and $\{\mathbf{h}_k^{V_i}\}_{k=1}^K$ are kept as *query* and *value*, respectively. Hence, we produce two different image-knowledge-infused embeddings denoted as $\mathbf{v}_i^{LV}$ and $\mathbf{v}_i^{VV}$ with $\{\mathbf{h}_k^{V_i}\}_{k=1}^K$ and $\{\mathbf{h}_k^{L_i}\}_{k=1}^K$ as *key*, respectively. The KAM outputs $\mathbf{v}_i^{LV}$ and $\mathbf{v}_i^{VV}$ as follows,

$$\begin{aligned} \mathbf{v}_i^{LV} &= \text{MHA}(\mathbf{e}_i, \{\mathbf{h}_k^{L_i}\}_{k=1}^K, \{\mathbf{h}_k^{V_i}\}_{k=1}^K), \\ \mathbf{v}_i^{VV} &= \text{MHA}(\mathbf{e}_i, \{\mathbf{h}_k^{V_i}\}_{k=1}^K, \{\mathbf{h}_k^{V_i}\}_{k=1}^K). \end{aligned} \tag{5}$$

Then, the outputs of the KAM will be combined with the input image embeddings to produce the augmented image embeddings ($\mathbf{e}_i^{'}$) as follows,

$$\mathbf{e}_i^{'} = \mathbf{e}_i + \mathbf{v}_i^L + \mathbf{v}_i^{LV} + \mathbf{v}_i^{VV}, \tag{6}$$

as shown in figure 3b. Then $\mathbf{e}_i^{'}$ is passed through the count decoder to produce the crowd count $\hat{c}_i$.

### 3.4 LOSS FUNCTION

At the pre-training stage of the image encoder, we use the pairwise ranking hinge loss ($L_r$) as described in equation 1. Then, to train the KAM and the count decoder, we utilize both $\mathcal{L}_1$ and $\mathcal{L}_2$ norm between the estimated and the ground truth count as follows,

$$
\begin{aligned}
\mathcal{L} &= \frac{1}{|\mathcal{B}|} \sum_{I_i \in \mathcal{B}} \left( \mathcal{L}_1(\hat{c}_i, c_i) + \lambda \, \mathcal{L}_2(\hat{c}_i, c_i) \right) \\
&= \frac{1}{|\mathcal{B}|} \sum_{I_i \in \mathcal{B}} \left( \|\hat{c}_i - c_i\|_1 + \lambda \, \|\hat{c}_i - c_i\|_2^2 \right),
\end{aligned}
\tag{7}
$$

where $\lambda$ is a hyperparameter that is set equal to $0.01$. In equation 7 $c_i$ and $\hat{c}_i$ are the ground truth and estimated count of the $I_i$ input image in the batch $\mathcal{B}$. The KAM and the count decoder are trained only with the remaining $20\%$ of simulated images that are not used to construct the reference database. Therefore, we readily have $c_i$ of each training image, since we know the ground truth for the simulated data.

### 4 EXPERIMENTAL DETAILS

#### 4.1 IMPLEMENTATION DETAILS

We use the Vision Transform (ViT-B/16) (Dosovitskiy et al., 2021) as the image encoder with pre-trained weights on ImageNet-21K (Deng et al., 2009) with the hidden dimension size set to 768. The input image size to the image encoder is $224 \times 224$. For the text encoder we use the Sentence Transformer (SentenceT) (Reimers & Gurevych, 2019). The SentenceT architecture has 6 Transformer block layers and outputs a 384 dimensional vector for each sentence. To reconcile the image embeddings with the text embeddings, we project the output of the image encoder from a 768 dimensional vector to a 384 dimensional vector. Moreover, we use an MLP for the count decoder to map the augmented image embeddings to the crowd count following Liang et al. (2022a).

We implement our framework with PyTorch (Paszke et al., 2019). All experiments are conducted on 4 NVIDIA RTX A6000 GPUs, and we use a batch size of 32 for pre-training the image encoder and training the KAM. First, the image encoder is trained for 200 epochs with unlabeled real crowd images and simulated crowd images. To pre-train the image encoder with the ranking loss, we use the AdamW optimizer (Loshchilov & Hutter, 2018) with a learning rate of 1e-3 and a weight decay of 0.01 factor and a linear warm-up over ten epochs. We use five ranked crops with a $1 : 0.75$ scaling ratio between consecutive crops on real crowd images following Liu et al. (2018), whereas we use four ranked crops with a $2 : 1$ scaling ratio for the simulated crowd images. During pre-training of the image encoder, we perform RandAugment (Cubuk et al., 2020), random horizontal flip, random Gaussian blur, and random color distortions. To train the KAM and the count decoder for generalized crowd-counting, we adopt the same optimizer with a learning rate of 1e-5 and perform training for 150 epochs using simulated data. To assess few-shot performance, we fine-tune the image encoder, KAM, and count decoder on real labeled crowd images.

#### 4.2 DATASETS AND METRICS

For the proposed method, we use the GCC dataset (Wang et al., 2019) to construct the reference dataset and to train the KAM and the count decoder. We evaluate the proposed method on five publicly available crowd datasets: JHU-Crowd++ (Sindagi et al., 2020), ShanghaiTech Part A and B (Zhang et al., 2016), UCF-QNRF (Idrees et al., 2018), and NWPU-Crowd (Wang et al., 2020). The performance is evaluated with the mean absolute error (MAE) and mean squared error (MSE).

Table 1: Crowd counting performance on JHU-Crowd++, UCF-QNRF, and ShanghaiTech-Part A and B datasets. We compare with other annotator-free methods and missing results are due to unavailable metrics in the corresponding paper. We provide the type of training data used by each method. The data domains are either **Re**al or **Si**mulated. Each method uses as **La**beled, **Un**labeled, and **Ps**eudo labeled data.

| Method | Venue | Training data | SHB MAE | SHB MSE | JHU MAE | JHU MSE | SHA MAE | SHA MSE | QNRF MAE | QNRF MSE |
|---|---|---|---|---|---|---|---|---|---|---|
| MCNN (Zhang et al., 2016) | CVPR'16 | Re La | 26.4 | 41.3 | 188.9 | 483.4 | 110.2 | 173.2 | 277.0 | 426.0 |
| P2PNet (Song et al., 2021) | ICCV'21 | Re La | 6.3 | 9.9 | - | - | 52.7 | 85.1 | 85.3 | 154.5 |
| CLTR (Liang et al., 2022b) | ECCV'22 | Re La | 6.5 | 10.6 | 59.5 | 240.6 | 56.9 | 95.2 | 85.8 | 141.3 |
| STEERER (Han et al., 2023) | ICCV'23 | Re La | 5.8 | 8.5 | 54.3 | 238.3 | 54.5 | 56.9 | 74.3 | 128.3 |
| GCC-SFCN (Wang et al., 2019) | CVPR'19 | Si La / Re Un | **19.9** | **28.3** | - | - | 123.4 | 193.4 | 230.4 | 384.5 |
| CSS-CCNN (Babu Sam et al., 2022) | ECCV'22 | Re Un | - | - | 217.6 | 651.3 | 197.3 | 295.9 | 437.0 | 722.3 |
| CrowdCLIP (Liang et al., 2023) | CVPR'23 | Re Ps | 69.3 | 85.8 | 213.7 | 576.1 | 146.1 | 236.3 | 283.3 | 488.7 |
| SYRAC (D'Alessandro et al., 2023) | arXiv | Re Ps | 49.0 | 60.3 | 194.0 | 583.9 | 196.0 | 295.2 | 390.0 | 697.5 |
| AFreeCA (D'Alessandro et al., 2024) | ECCV'24 | Re Ps | 35.0 | 50.7 | 173.8 | 519.4 | 152.7 | 219.0 | 283.1 | 453.2 |
| Ours | | Si La / Re Un | 23.0 | 30.7 | **142.3** | **443.6** | **118.4** | **186.1** | **214.9** | **363.4** |

Table 2: Performance on the NWPU-Crowd test dataset. We used the publicly available code bases to evaluate the performance of the annotator-free methods. We provide the type of training data used by each method. The data domains are either **Re**al or **Si**mulated. Each method uses as **La**beled, **Un**labeled, and **Ps**eudo labeled data.

| Method | Venue | Training data | Overall MAE | Overall MSE | Scene Level (MAE) Avg. | S0 | S1 | S2 | S3 | S4 |
|---|---|---|---|---|---|---|---|---|---|---|
| MCNN (Zhang et al., 2016) | CVPR'16 | Re La | 232.5 | 714.6 | 1171.9 | 356.0 | 72.1 | 103.5 | 509.5 | 4818.2 |
| P2PNet (Song et al., 2021) | ICCV'21 | Re La | 72.6 | 331.6 | 510.0 | 34.7 | 11.3 | 31.5 | 161.0 | 2311.6 |
| CLTR (Liang et al., 2022b) | ECCV'22 | Re La | 74.4 | 333.8 | 532.4 | 4.2 | 7.3 | 30.3 | 185.5 | 2434.8 |
| STEERER (Han et al., 2023) | ICCV'23 | Re La | 63.7 | 309.8 | 410.6 | 48.3 | 6.0 | 25.9 | 158.3 | 1814.5 |
| CSS-CCNN (Babu Sam et al., 2022) | ECCV'22 | Re Un | 433.0 | 868.3 | 1965.3 | 368.3 | 233.7 | 289.6 | 689.6 | 8245.3 |
| CrowdCLIP (Liang et al., 2023) | CVPR'23 | Re Ps | 374.9 | 899.4 | 1646.2 | 305.7 | 190.5 | 237.3 | 677.2 | 6820.6 |
| SYRAC (D'Alessandro et al., 2023) | arXiv | Re Ps | 344.5 | 958.5 | 1540.6 | 268.5 | 182.9 | **215.4** | **610.8** | 6425.8 |
| Ours | | Si La / Re Un | **340.1** | **863.8** | **1358.0** | **248.9** | **153.3** | 226.4 | 672.1 | **5489.7** |

# 5 RESULTS AND ANALYSIS

## 5.1 ANNOTATOR-FREE PERFORMANCE

As reported in tables 1 and 2, the proposed ReGe-Count surpasses state-of-the-art methods: GCC-SFCN, CrowdCLIP, AFreeCA by considerable margins across all evaluated datasets. Moreover, ReGe-Count surpasses state-of-the-art annotator-free methods by considerable margins in terms of MAE for the NWPU-Crowd test dataset. The performance against CSS-CCNN comes from performing zero-shot on the target distribution under weak supervision, which has been better than self-supervision. Then, actively using language information has aided in surpassing CrowdCLIP, which does not use text information for estimation. The performance across different datasets indicates that the proposed method performs well under different conditions, as these datasets specifically represent congested and sparse scenes. Furthermore, we have provided some qualitative results in figure 5 with the individual patch counts to better illustrate the performance of our method. Furthermore, ReGe-Count method demonstrates highly competitive performance against some widely adopted fully supervised methods like MCNN (Zhang et al., 2016).

## 5.2 ABLATION STUDY

**Effectiveness of knowledge retrieval.** In figure 4, we provide the top-4 retrieved simulated samples and the corresponding text information for two query images. The first query image is an indoor photo where the individuals are placed in an ordered manner. The first image retrieved by the query resembles the ordered structure in the image, even though the retrieved patch is an outdoor image (the GCC dataset contains only outdoor images.) Though the rest of the samples do not contain the ordered structure, those images mimic other aspects like the orientation of how individuals are placed and size constancy. To quantify the spatial similarity, we considered the density maps.

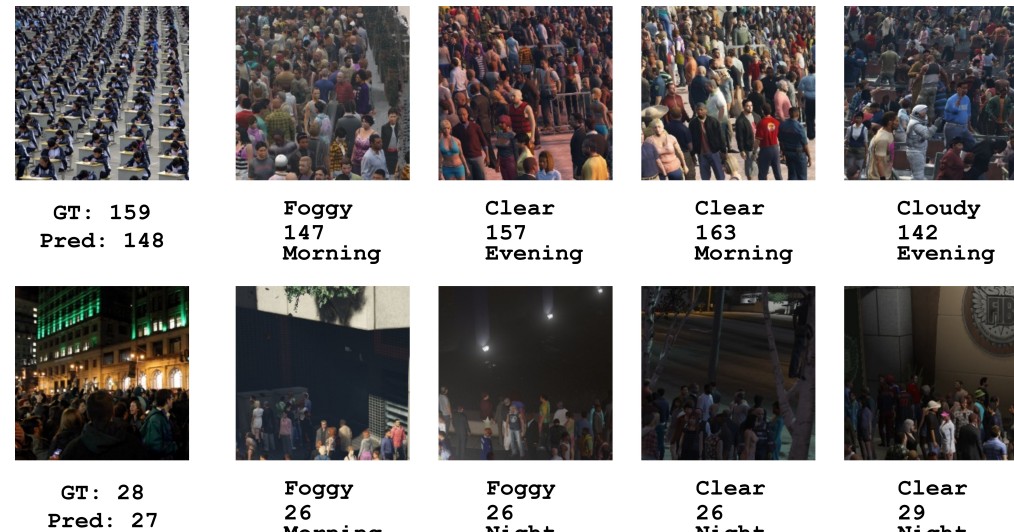

Figure 4: Top-4 retrieved simulated samples and text descriptions for a real query patch. The first query is an indoor image with an ordered placement of people. The retrieved patches resemble the ordered structure of the query, though retrieved images are outdoor. The second query is an outdoor night image. The retrieved samples match the image conditions and the orientation of the location.

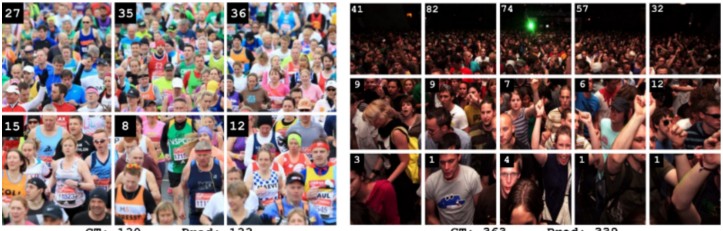

Figure 5: Qualitative results from ReGe-Count.

We measured the SSIM of each retrieved image with the query image, where the SSIM values varied between 0.88 and 0.82, which indicates a high spatial similarity of object placement. The second query image is a dark and outdoor sample with random placement. The re-trieved samples for the second query either have a darker background or belong to the night conditions. Specifically, the first extracted sample also simulates the background of the query image. Like in the first query image, the orientation of human placement is also present in the extracted simulated patches. Also, the SSIM values for the density maps for the second example range from 0.90-0.92, indicating a strong spatial similarity for object placement. However, this high SSIM number could also be driven by the fact that there are fewer people in the second case, and most of the density map is empty. Regardless, figure 4 gives insight as to how retrieving from an external dataset can facili-tate the generalized capabilities of the network as the crowd count of the extracted patches provides closer estimates.

To validate the effectiveness of our proposed method for annotator-free crowd counting on the target distribution, we compare ReGe-Count with CrowdCLIP, CSS-CCNN, and DGCC (Du et al., 2023). Note that Crowd-CLIP is a vision-language-based counting method, CSS-CCNN is a self-supervised counting method, and DGCC is a domain generalization-based counting method. Here, we only consider labeled simulated data and unlabeled real data and follow the training pipeline provided in public codebases. For DGCC, we train the pipeline with only simu-lated data, as DGCC requires labeled data. However, for CrowdCLIP and CSS-CCNN, we pre-train

Table 3: Generalized cross-dataset performance comparison with simulated GCC dataset training.

| Method | JHU | SHA | QNRF | SHB |
|---|---|---|---|---|
| DGCC | 544.4 | 351.2 | 454.8 | 112.1 |
| CSS-CCNN | 234.3 | 258.8 | 315.5 | 77.0 |
| CrowdCLIP | 226.7 | 162.7 | 325.4 | 82.9 |
| Ours* | 170.2 | 143.1 | 223.6 | 28.6 |
| Ours | 142.3 | 118.4 | 214.9 | 23.0 |

Ours* does not have the retrieval module and count decoder is fine-tuned with labeled synthetic data.

the image encoders using both simulated and real crowd datasets, as the pre-training stage only requires unlabeled images. Also, in the real crowd dataset for pre-training, we don't include the training images of the test distribution in comparison with ReGe-Count training scheme. The results of this ablation are provided in table 3. As observed from table 3, DGCC fails to generalize to real crowd images when purely trained on simulated crowd images. Furthermore, CrowdCLIP and CSS-CCNN performed worse than ReGe-Count by a significant margin. This is because CrowdCLIP operates in the classification scenario instead of our regression-based method. Also, in contrast to our ReGe-Count, CrowdCLIP does not use language understanding to produce the crowd count. Note that CSS-CCNN assumes the crowd counts distribution of patches to follow the power law for simulated images, which might also be invalid. Furthermore, we consider the performance without the KAM in Ours*, which performs worse than the proposed method. We provide an analogy for this observation. In the training stage of Ours*, the network learns multi-modal (vision-language) concepts with the training distribution. Then, given a query image during testing, Ours* attempts to perform a closed-book inference and may return a false prediction if it cannot relate the query (real image) with the learned concepts.

**Qualitative analysis.** To understand the most influential aspects of the pipeline, especially in the Knowledge Augmentation Module, we consider the attention weights by different augmentations. First, we consider the attention maps (see figure 6) in the KAM module for a test image and the closest retrieved image for different keywords in the text description. In both cases, the maps corresponding to the 'count' keyword have high scores compared to the maps of the other keywords. This indicates that the 'count' text features will highly influence the augmented embeddings passed to the decoder. Further, the attention maps highlight the areas of crowds that exist in the scene for the retrieved image, demonstrating the visual understanding of people with the count. In addition, the 'time of day' keyword has provided some background context in the retrieved scene in the first example, whereas the 'weather' keyword features will have a minimal effect on the augmented embeddings compared to the other two. Also, we consider the attention between the test image's image embeddings and the retrieved description's text embeddings. By averaging and normalizing the attention weights, we could compute attention scores assigned to each word token. For instance, we considered an image crop of an indoor scene with low illumination. For this example, the attention scores produced for each word token were: The (**0.000**) image (**0.002**) has (**0.000**) a (**0.000**) clear (**0.030**) weather (**0.000**) with (**0.000**) 52 (**0.905**) people (**0.003**) in (**0.000**) the (**0.000**) night (**0.060**). The attention scores are high for the crowd count, time of day, and weather, as these three keywords carry information among different images because the remaining text words are common across all text descriptions. Since both test image and retrieval embeddings are generated from the same encoder, these attention scores highlight which feature maps are more influential due to the way the attention mechanism is developed.

**Effective use of text modality.** We compare the change in performance with different text prompts to demonstrate the effective use of text modality for crowd-counting in ReGe-Count compared to CrowdCLIP. While CrowdCLIP has explored text modality for crowd counting first, the setting proposed in CrowdCLIP uses text embeddings as reference vectors to train the image encoder rather than using text information to produce the count. For example, when we change the text prompt from "The photo contains [**count**] people" to "There are [**count**] people in the photo", the performance of CrowdCLIP changed significantly (283.3→488.1) as opposed to ours (214.9→216.4). Hence, CrowdCLIP has underutilized the potential of text information compared to our work.

Table 4: Different augmentations

| Fusion type | | | $K$ | MAE | | |
|---|---|---|---|---|---|---|
| $\mathbf{v}_i^L$ | $\mathbf{v}_i^{LV}$ | $\mathbf{v}_i^{VV}$ | | JHU | SHA | QNRF |
| ✗ | ✗ | ✗ | - | 170.2 | 143.1 | 223.6 |
| ✓ | ✗ | ✗ | 32 | 148.6 | 123.6 | 224.4 |
| ✓ | ✓ | ✓ | 32 | 142.3 | 118.4 | 214.9 |
| ✓ | ✓ | ✓ | 16 | 145.9 | 121.4 | 220.4 |
| ✓ | ✓ | ✓ | 64 | 143.9 | 119.7 | 217.3 |

Table 5: Ablation of keywords

| Method | JHU | SHA | QNRF |
|---|---|---|---|
| baseline | 142.3 | 118.4 | 214.9 |
| count | 147.5 | 117.6 | 215.8 |
| + time of day | 143.5 | 118.1 | 215.1 |
| + weather | 147.8 | 118.3 | 216.1 |

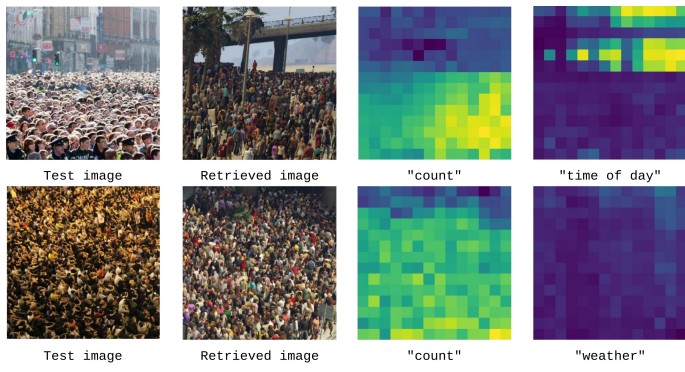

Figure 6: Attention maps corresponding to different keywords

**Different augmentation architectures.** We conducted additional experiments to assess various design options within the KAM architecture, as outlined in table 4 since the KAM is the sole addition to the baseline architecture. For the baseline, the count decoder is trained with the simulated data without any image-text retrieval and knowledge enhancement. Initially, we consider $\mathbf{v}_i^L$ as the ultimate augmented representation. In this scenario, the KAM assimilates relevant information from the reference texts, generating the final embedding. Including textual cues leads to a notable performance improvement compared to the baseline. After that, we augment $\mathbf{v}_i^{LV}$ and $\mathbf{v}_i^{VV}$, separately. Providing visual cues based on the image-text relation has not seemed effective. Still, it has improved the performance when combined with the remaining augmentations. More details are provided in the supplementary.

**Different $K$-value in retrieval.** We experiment with different retrieval quantities and their effect on performance. We vary between $K = \{16, 32, 64\}$ for image and text pairs from the external simulated dataset. The results are tabulated from rows 4-6 in table 4. Our module exhibits consistent performance across different K values, with the model achieving slightly superior results when K is set to 32 compared to other configurations. The performance decrease with a higher $K$ value could arise when the count information provided by the least similar retrieved crops is significantly different from the true count.

**Effect of the text description.** The text description adds context to retrieved scenes, but its impact on performance varies. Ablation studies in table 5 show that using all three keywords produces similar results for SHA and QNRF datasets, indicating that additional context keywords do not significantly influence performance. However, for the JHU dataset, the *time of day* keyword improves performance, unlike the *weather* keyword, which can be omitted without affecting results. The difference arises because JHU, a larger dataset, includes diverse scene illuminations, while SHA and QNRF primarily feature bright scenes. Thus, the *time of day* keyword enhances context for JHU by differentiating illumination levels, whereas it has little effect on the other datasets.

**Few-shot performance.** We analyze the few-shot performance of crowd counting with knowledge retrieval. Here, we fine-tune the pre-trained image encoder for a fair comparison with weakly-supervised TransCrowd (Liang et al., 2022a). The few shot performance (MAE) of ReGe-Count is tabulated in table 6 for the JHU-Crowd++, ShanghaiTech, and UCF-QNRF datasets. Values reported in table 6 are the average of five realizations for each training data percentage. ReGe-Count delivers state-of-the-art counting results for weakly supervised methods surpassing TransCrowd while operating at 90% of the train data.

## 6 CONCLUSION

ReGe-Count introduces a novel framework for transferring language knowledge to enhance generalized crowd counting. It is the first to apply knowledge retrieval to improve annotator-free crowd-counting accuracy. Notably, ReGe-Count achieves state-of-the-art performance in annotator-free crowd counting and addresses the high annotation costs associated with labeling real crowd images. By effectively leveraging large-scale, annotation-free simulated data, our approach underscores the potential of knowledge-enhanced models for crowd counting, paving the way for future research at the intersection of vision and language models.

Table 6: Few-shot and full training performance with knowledge retrieval.

| Method | JHU | SHA | QNRF | SHB |
|---|---|---|---|---|
| 10%-Real | 95.0 | 158.3 | 212.3 | 22.3 |
| 25%-Real | 82.7 | 138.2 | 183.2 | 20.3 |
| 50%-Real | 67.7 | 107.1 | 152.9 | 14.9 |
| 90%-Real | 55.2 | 64.8 | 95.9 | 9.2 |
| TransCrowd | 56.8 | 66.1 | 97.2 | 9.3 |

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

APPENDIX

This appendix is organized as follows.

- In section A, we illustrate the simulated image embedding retrieval process, including the intermediate processes.

- In section B, we provide results and explanations for additional ablation studies.

- In section C, examples of the retrieved simulated samples in the case of negative samples and congested scenes.

- In section D, we compare the inference performance of the proposed method with other annotator-free crowd counting methods.

- In section E, we provide details of the datasets and metrics we used.

- In section F, the computational efficiency of the image retrieval process is analyzed.

- In section G, the computational cost and inference performance against counting performance are discussed.

- In section H, we provide a theoretical explanation for the improvement from the knowledge augmentation.

## A  IMAGE RETRIEVAL PROCESS

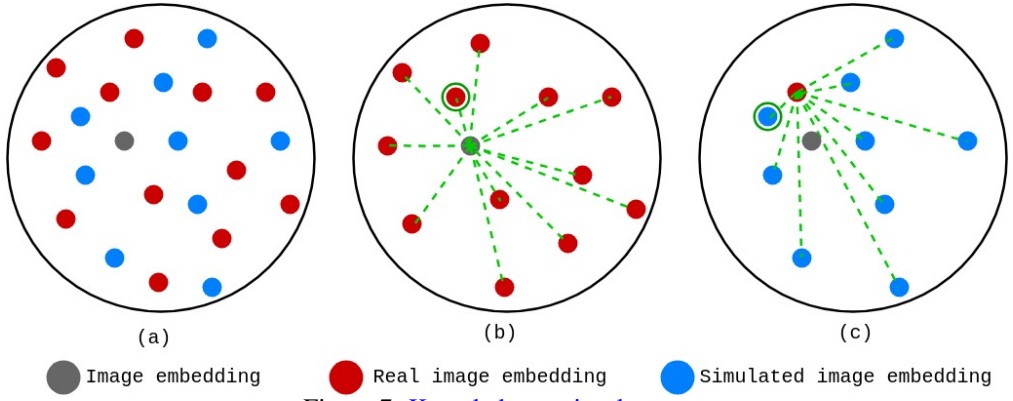

Figure 7: Knowledge retrieval process

In this section, we elaborate on the knowledge retrieval process described in section 3.2 using illustrations. In knowledge retrieval, we extract $K$ image-text pairs for an input image $I_i$. First, we get the image embeddings $\mathbf{e}_i$ of $I_i$ using the pre-trained image encoder ($\Psi$). Since $\mathbf{e}_i$ is produced from $\Psi$ and $\Psi$ is trained using real images and simulated images, we assume $\mathbf{e}_i$ lies on the same embeddings space as the image embeddings of the real and simulated images. This is demonstrated in figure 7a where the gray color image embedding is in the same manifold as the red color real image embeddings and blue color simulated image embeddings. Next, we perform the maximum inner product search (MIPS) with the image embeddings of the real crowd image crops in the reference database.

In figure 7b, we demonstrate the retrieval of the closest embedding. In MIPS, first, we compute the distance between $\mathbf{e}_i$ and real image embeddings under the vector inner product. Then, we find the real image embedding closest to or the most similar to $\mathbf{e}_i$. Then, we perform MIPS between the selected real image embedding and the simulated image embeddings. In figure 7c, we demonstrate the retrieval of the closest simulated embedding.

Furthermore, in figure 8, we provide an illustration of the two-stage retrieval process with examples for the 2-nearest neighbors. First, the input image embedding will perform MIPS to find the closest embeddings from the real image dataset. Then, for each real image embedding (outlined in red), MIPS will find the closest embeddings from the simulated image dataset. These simulated image embeddings (outlined in blue) are passed to the KAM for knowledge augmentation.

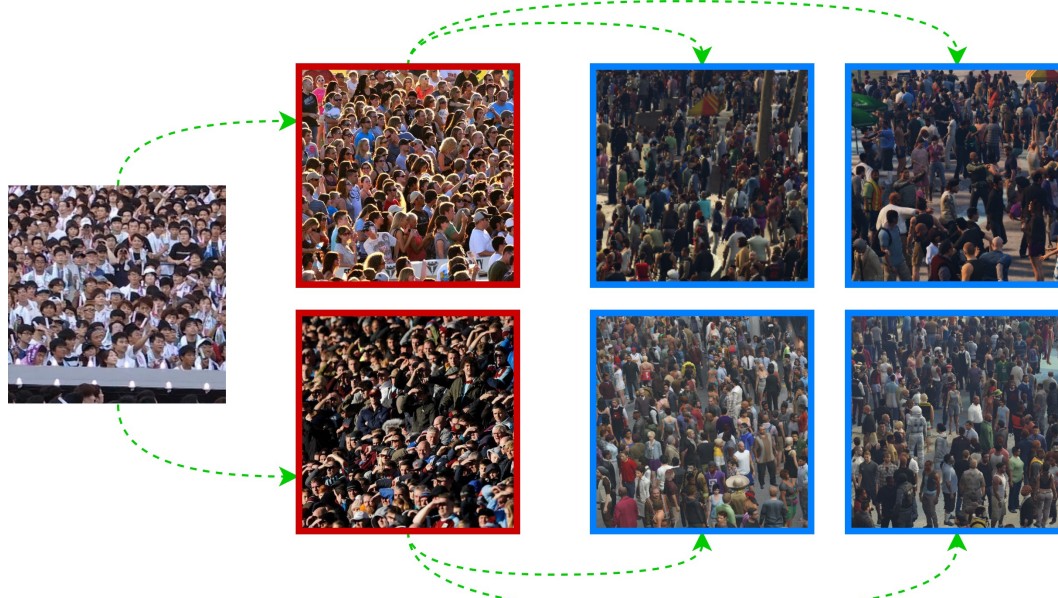

Figure 8: Two-stage retrieval with examples with 2-nearest neighbors. First, we find the nearest real image embeddings (outlined in red). Then, for each real image embedding we find the nearest simulated image embeddings (outlined in blue).

## B  ADDITIONAL ABLATION STUDIES

**Detailed ablation on augmentations** The crucial module of the pipeline is the Knowledge Augmentation Module (KAM). In the KAM, we use three different embedding augmentations, as depicted by the first three columns of table 4.

The performance gain by each augmentation type is provided for only $\mathbf{v}_i^L$ in table 4. Therefore, to understand which augmentation types improve the performance, we provide the counting performance for each augmentation type and their combinations compared against the baseline performance in the table 7. The most performance gain has come from the components $\mathbf{v}_i^L$ and $\mathbf{v}_i^{VV}$ compared to the baseline method. The two augmentations deliver text information and visual information, respectively, but the cross-attention is taken between the image embeddings and retrieved patch embeddings. However, the performance gain from $\mathbf{v}_i^{LV}$ is marginal compared to the other two augmentations where the cross-attention is taken between the image and retrieved text embeddings.

Table 7: Detailed ablation of different augmentations

| Fusion type | | | $K$ | MAE | | |
|---|---|---|---|---|---|---|
| $\mathbf{v}_i^L$ | $\mathbf{v}_i^{LV}$ | $\mathbf{v}_i^{VV}$ | | JHU | SHA | QNRF |
| ✗ | ✗ | ✗ | - | 170.2 | 143.1 | 223.6 |
| ✓ | ✗ | ✗ | 32 | 148.6 | 123.6 | 215.4 |
| ✗ | ✓ | ✗ | 32 | 152.8 | 128.3 | 219.8 |
| ✗ | ✗ | ✓ | 32 | 145.3 | 125.8 | 216.5 |
| ✓ | ✓ | ✗ | 32 | 149.3 | 122.3 | 221.7 |
| ✓ | ✗ | ✓ | 32 | 142.8 | 118.8 | 215.7 |
| ✓ | ✓ | ✓ | 32 | 142.3 | 118.4 | 214.9 |

**Different retrieval processes.** We consider the effect of not using real crowd images in the retrieval process and directly retrieving from the simulated dataset. However, the image encoder is pre-trained with real crowd images in the mix. When directly retrieving from the simulated dataset, we observed an MAE of 243.8 for JHU-Crowd++, which is poorer than the CrowdCLIP and CSS-CCNN performances. This is because, even though the image encoder is trained to embed real and simulated images in the same space, the training of the KAM and the decoder disregards the domain

gap between real and simulated images.

**Different amount of reference data.** We evaluate the effect of the reference set size on the performance for five cardinalities by randomly sampling $10\%, 25\%, 50\%, 75\%$, and $80\%$ image-text pairs from GCC dataset. The ablation study recorded an average MAE of 224.2, 210.6, 174.0, 144.7, and 142.8, respectively, on JHU-Crowd++ for five trials. The performance was higher for larger reference set sizes. This is because in larger reference databases, for a given test image crop, a positive simulated crop is closer than in smaller databases, providing more accurate information retrieval.

## C   QUALITATIVE RESULTS

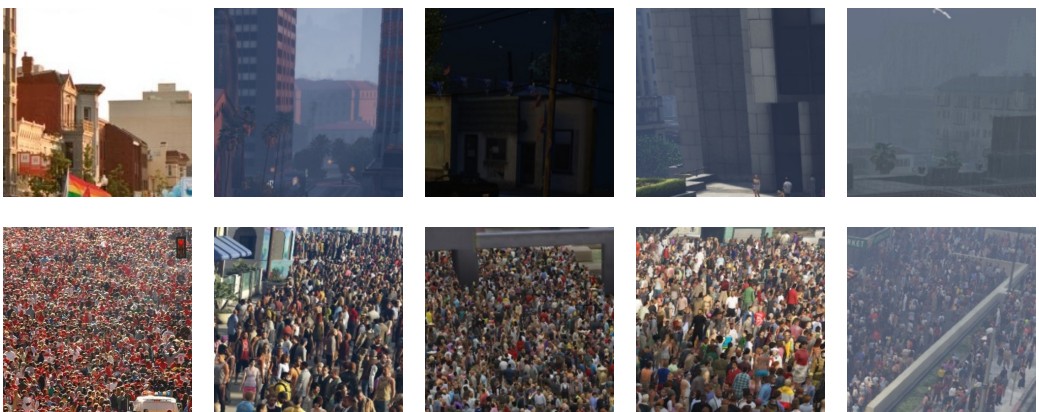

Figure 9: Retrieved synthetic crops for a negative sample (top row) and congested sample (bottom row).

We provide qualitative results to demonstrate the performance of the retrieval process in the proposed method in figure 9. In the first row, we have a negative test image. Most of the retrieved test images for the negative sample had zero crowd counts and had similar backgrounds. Nonetheless, some retrieved patches had smaller counts ($< 4$) where the background was similar to the test image. In the second row, we have a congested test image. The retrieved patches for the congested scene are of similar crowd density patterns, even though most of the images do not fill up the entire image. This validates the idea that the retriever searches for simulated images that resemble the crowd density pattern of the test image, as first mentioned in section 5.2 with figure 4.

## D   INFERENCE SPEED

We present a comparison of inference speeds, as outlined in table 8. The runtime of our proposed annotator-free method is significantly higher than other annotator-free methods, such as CrowdCLIP and CSS-CCNN. CrowdCLIP gives an interval of FPS values as it uses progressive filtering for crowd patches with people, whereas the proposed work only has one forward pass through the image encoder. Then, CSS-CCNN utilizes a larger decoder to estimate the density map to predict the count, whereas we only use a linear layer to estimate the count directly. Additionally, the use of a vector database to retrieve samples improves inference time as the retrieval operation is simply the vector inner product. Notably, fully supervised methods necessitate maintaining high-resolution features to produce quality density maps. For instance, in CSRNet Li et al. (2018), features are 1/8 the size of the input, while in BL Ma et al. (2019), they are 1/16 the size, resulting in slower inference speeds.

## E   DATASETS

**JHU-Crowd++**Sindagi et al. (2020) contains $2,722$ training images, $500$ validation images, and $1,600$ testing images, collected from diverse scenarios. The total number of people in each image ranges from 0 to $25,791$.

**ShanghaiTech**Zhang et al. (2016) contains $1,198$ crowd images with $330,165$ annotations. The images of the dataset are divided into two parts: Part A and Part B. In particular, Part A contains 300

Table 8: The comparisons of Frames Per Second (FPS) between our method and other methods. The results are conducted on an NVIDIA A6000 GPU

| Method | Annotated data | Label | Resolution | FPS |
|---|---|---|---|---|
| CSRNet Li et al. (2018) | Real | density | $1024 \times 768$ | 18.4 |
| BL Ma et al. (2019) | Real | density | $1024 \times 768$ | 21.3 |
| CSS-CCNN Babu Sam et al. (2022) | ✗ | ✗ | $1024 \times 768$ | 37.4 |
| CrowdCLIP Liang et al. (2023) | Real | count text | $1024 \times 768$ | [24.0, 50.8] |
| Ours | Synthetic | count | $1024 \times 768$ | 42.8 |

training images and 182 testing images, and Part B consists of 400 training images and 316 testing images.

**UCF-QNRF** Idrees et al. (2018) contains $1,535$ images captured from unconstrained crowd scenes with about one million annotations. It has a count range of 49 to $12,865$, with an average count of $815.4$. Specifically, the training set consists of $1,201$ images and the testing set consists of 334 images.

**NWPU-Crowd** Wang et al. (2020), a large-scale and challenging dataset, consists of $5,109$ images, $2,133,375$ instances annotated elaborately. To be specific, the images are randomly split into three parts, including training, validation, and testing sets, which contain $3,109$, 500, and $1,500$ images, respectively.

**GCC** Wang et al. (2019) dataset consists of $15,212$ images, with a resolution of $1080 \times 1920$, containing $7,625,843$ persons. Compared with the existing datasets, GCC is a larger-scale crowd counting dataset in terms of both the number of images and the number of persons.

**Metrics** we used for evaluate the counting performance were MAE and MSE as defined below:

$$\text{MAE} = \sum_{n=1}^{N} \frac{1}{N} |c_n - \hat{c}_n| \text{ and MSE} = \sqrt{\sum_{n=1}^{N} \frac{1}{N} |c_n - \hat{c}_n|^2}, \tag{8}$$

where $c_n$ and $\hat{c}_n$ are the groundtruth and predicted crowd count of the $n^{\text{th}}$ image out the the $N$ images tested.

## F  EFFICIENCY ANALYSIS

For the retrieval process, we use the naive maximum inner product search. This involves computing the similarity between image embeddings and crop embeddings in the reference database and sorting the similarity scores to find the closest neighbors.

Suppose the reference database is of size $N$, the embedding dimensionality is of size $d$, and we need to find the nearest $k$ neighbors. Then, the computational efficacy of the whole process is $O(N \cdot d + N \cdot \log k)$. Accordingly, as the retrieval space scales, the time it takes for the retrieval process will increase. However, for larger reference databases, using approximation methods such as the k-d tree, the computational complexity can be reduced to $O(\log N)$ for smaller dimensional sizes, but still, the time consumed will increase with the size of the reference database.

## G  COMPUTATIONAL COST AND COMPLEXITY

We provide a comparison for the inference speed in table 8 in supplementary material. However, we will itemize the inference time and the computational complexity for the model with and without the KAM, along with the accuracy. For the proposed method, the inference time and computational complexity are influenced by three components: Image encoder and count decoder, knowledge retrieval process, and KAM. We tabulate the computational complexity in the following table.

The MAE performance for the JHU public dataset is given in the table 9. The baseline corresponds to the network without the KAM and the minimum model latency without the proposed improvements. The MIPS corresponds to the retrieval process with the inner product search to find the 16 nearest neighbors for a given image embedding. In table 9, GFLOPS measures the rate at which a computing system can execute floating-point operations. This rate is influenced by the number of retrieved data we feed into the KAM module. Which is why the GLOPS is higher than the baseline. However, for a single retrieved image-text pair, the GLOPS is 4.368. As a fellow transformer-based

Table 9: Computational efficiency of the architecture

|          | GFLOPS  | Time (ms) | MAE   |
|----------|---------|-----------|-------|
| Baseline | 70.564  | 8.55      | 170.2 |
| MIPS     | -       | 3.51      | -     |
| KAM      | 151.196 | 18.32     | 142.3 |

model, crowd clip has a higher number of transformer modules compared to our architecture since CrowdCLIP uses two ViT-B/16 for visual encoding in addition to the transformer-based text encoder.

## H  THEORETICAL ANALYSIS

To explain the contribution of knowledge augmentation to improving zero-shot crowd-counting, we use a probabilistic approach.

The goal is to predict the crowd-count $c_i$ for the target embedding $\mathbf{e}_i$. Using a probabilistic framework, the prediction can be expressed as:

$$P_{source}(c_i|\mathbf{e}_i) = P_{source}(c_i|\mathbf{e}_i^{'}),$$

where the augmented embedding is:

$$\mathbf{e}_i^{'} = \mathbf{e}_i + \mathbf{v}_i^{L} + \mathbf{v}_i^{LV} + \mathbf{v}_i^{VV}.$$

Using Bayes' rule, we can rewrite the probability as follows:

$$P_{source}(c_i|\mathbf{e}_i^{'}) \propto P(\mathbf{e}_i^{'}|c_i)P_{source}(c_i),$$

where $P(\mathbf{e}_i^{'}|c_i)$ and $P(c_i)$ denote the likelihood of the augmented embedding given the count and the prior probability of the count derived from the source distribution.

Then, the likelihood can be decomposed as

$$P(\mathbf{e}_i^{'}|c_i) \propto P(\mathbf{e}_i|c_i) \prod_{n} P(\mathbf{v}_i^{n}|c_i)$$

where $\mathbf{v}_i^{n}$ is each individual augmentation type from the KAM. However, each individual augmentation is computed from the KAM using the retrieved embeddings from the reference database. Therefore, the likelihood can be updated as:

$$P(\mathbf{e}_i^{'}|c_i) \propto P(\mathbf{e}_i|c_i) \prod_{n} \prod_{k=1}^{K} P(r_{ik}^{n}|c_i)$$

where $r_{ik}^{n}$ denotes the retrieved embedding augmented with multi-head attention (MHA), and $k$ is the index of the retrieved embedding. Each $\mathbf{v}_i^{n}$ thus encodes the aggregated likelihood information from its corresponding patches, ensuring that $\mathbf{e}_{\text{aug}}$ effectively aligns with the count $c_i$ as MHA behaves as a projection of the query embedding to the key embedding. Consequently, the retrieved embeddings $\mathbf{v}_i^{n}$ encode domain-specific patterns, improving the likelihood estimation.

Without the retrieved embeddings, the likelihood distribution will only depend on $\mathbf{e}_i$, and as augmentations are introduced, the likelihood distribution is influenced by the source domain information. The influence of the source likelihood increases with the number of retrieved embeddings. In return, the posterior distribution $P_{source}(c_i|\mathbf{e}_i^{'})$ becomes a sharper posterior distribution. As the posterior distribution becomes sharper, the uncertainty involved with the prediction reduces, improving the prediction accuracy.