# OpenReview forum: "Retrieval-based Zero-shot Crowd Counting"
_ICLR.cc/2025/Conference — Submitted to ICLR 2025_

### Official Review · Reviewer_XkFU · 2024-10-23

**Soundness:** 2
**Presentation:** 3
**Contribution:** 3
**Rating:** 5
**Confidence:** 4

**Summary:**

The paper titled "Retrieval-Based Zero-Shot Crowd Counting" introduces a novel approach named ReZeS-Count that leverages knowledge retrieval to enhance zero-shot crowd counting performance.  The method addresses the challenge of distribution gap between real and simulated data by retrieving simulated crowd images and their text descriptions to augment the image embeddings of real crowd images.  This is achieved through a pre-training and retrieval mechanism that incorporates unlabeled real crowd images along with simulated data, thus reducing the annotation cost associated with real datasets.

**Strengths:**

#### Originality
- Introduces the ReZeS-Count framework, innovatively combining knowledge retrieval with zero-shot crowd counting, utilizing simulated data and text information to enhance real image embeddings.

#### Quality
- Demonstrates the effectiveness of the proposed method through extensive experiments on five public datasets, surpassing current crowd counting approaches.

#### Clarity
- The paper is well-structured, with a detailed explanation of the framework and its components in the methodology section.

#### Significance
- Addresses the challenges of annotation costs and distribution differences between real and simulated data in crowd counting.

**Weaknesses:**

1. While the paper claims state-of-the-art results on five public datasets, it would benefit from testing on more diverse datasets, particularly those with varying densities and complexities, to further validate the robustness of the ReZeS-Count framework.

2. Ablation Studies: The paper could provide more detailed ablation studies to isolate the contribution of each component of the framework.  For instance, the impact of the knowledge retrieval module could be quantified independently to understand its specific contribution to the overall performance.

3. Model Complexity: The paper could provide more insight into the computational complexity of the ReZeS-Count framework. Understanding the trade-offs between accuracy and computational resources is critical for practical applications.

4. Comparison with State-of-the-Art: While the paper claims state-of-the-art performance, but the comparison method is not the latest.

5. Theoretical Foundations: The paper could benefit from a deeper theoretical analysis of why the knowledge retrieval approach works for zero-shot crowd counting.

**Questions:**

Please refer to Section Weaknesses.

---

> ### Author Response · Authors · 2024-11-23
> **Initial response to Reviewer XkFU: part 1**
>
> Thank you for your thoughtful and constructive feedback on our manuscript. We appreciate your suggestions to test on more diverse datasets, provide detailed ablation studies, analyze computational complexity, compare with the latest state-of-the-art methods, and delve deeper into the theoretical foundations of our approach.
>
> **Q1. Crowd-counting Benchmarks**
> The public datasets used in the paper are prominently used in the field of crowd-counting. Furthermore, these five datasets are also used by other ground-truth annotation-free methods such as CrowdCLIP [3], CSS-CCNN [1], AFreeCA [2], which also only use the above public benchmarks.
>
> Furthermore, we provide the performance comparison for the NWPU dataset in `table 2`, which is a public evaluation benchmark that has not released the test ground truth. The NWPU benchmark evaluates the complexity of different scenes and crowd density. We provide the performance comparison for different scene levels for the NWPU dataset in `table 2`, and the scene level index is proportional to the crowd density of the scene (e.g., S1 is low crowd density, and S5 is high crowd density). Therefore, we believe the amount of datasets presented in this paper is sufficient to convey the robustness and contribution of the proposed work.
>
> ---
>
> **Q2. Detailed Ablation Studies**
> In the proposed work, the pipeline is comprised of three main components: image encoder, Knowledge Augmentation Module (KAM), and count decoder. The key component of the proposed framework is the KAM. Furthermore, the KAM module has three augmentations: $\mathbf{v}_i^L$, $\mathbf{v}_i^{LV}$, and $\mathbf{v}_i^{VV}$. Therefore, we could consider the contribution of each individual augmentation type to the overall performance as well as the KAM as a whole unit. This performance boost of each item should be measured against a baseline where none of the augmentations is used.
>
> We have already given the performance for the baseline method in row 4 of `table 3`, where the image encoder is pre-trained, as explained in `section 3.1`, and the count decoder is fine-tuned with simulated data without using the retrieval process. Then, the contribution from the KAM to the overall performance is given in row 5 of `table 4`, which quantifies the improvement of the proposed KAM. Additionally, we have provided the performance gain from using the augmentation $\mathbf{v}_i^L$ in `table 4`, along with other augmentation schemes. However, in the following table, we tabulate the performance of each individual augmentation type and the KAM module against the baseline method.
>
> **Table A**: Ablation of augmentation schemes.
>
> | Augmentation     | JHU   | SHA   | QNRF  |
> |------------------|-------|-------|-------|
> | Baseline         | 170.2 | 143.1 | 223.6 |
> | $\mathbf{v}_i^L$ | 148.6  | 123.6   | 215.4 |
> | $\mathbf{v}_i^{LV}$ | 152.8  | 128.3   | 219.8 |
> | $\mathbf{v}_i^{VV}$ | 145.3  | 125.8   | 216.5 |
> | KAM              | 142.3 | 118.4 | 214.9 |
>
> ---
>
> **Q3. Computational Complexity Analysis**
> We provide a comparison for the inference speed in Table 6 in the supplementary material. However, we will itemize the inference time and the computational complexity for the model with and without the KAM, along with the accuracy. For the proposed method, the inference time and computational complexity are influenced by three components: image encoder and count decoder, knowledge retrieval process, and KAM.
>
> **Table B**: Computational complexity and inference time for different components.
>
> | Component        | GFLOPS   | Time (ms) | MAE     |
> |------------------|----------|-----------|---------|
> | Baseline         | 70.564  | 8.55     | 170.2   |
> | MIPS             |  -  | 3.51      | -       |
> | KAM              | 151.196    | 18.32      | 142.3   |
>
> The MAE performance for the JHU public dataset is given in the above table. The baseline corresponds to the network without the KAM and the minimum model latency without the proposed improvements. The MIPS corresponds to the retrieval process with the inner product search to find the 32 nearest neighbors for a given image embedding.
>
> ---
>
> **References**
> [1] Deepak Babu Sam, Abhinav Agarwalla, Jimmy Joseph, Vishwanath A Sindagi, R Venkatesh Babu, and Vishal M Patel. *Completely self-supervised crowd counting via distribution matching*. European Conference on Computer Vision, pp. 186–204, 2022.
> [2] Adriano D’Alessandro, Ali Mahdavi-Amiri, and Ghassan Hamarneh. *Afreeca: Annotation-free counting for all*. arXiv preprint arXiv:2403.04943, 2024.
> [3] Dingkang Liang, Jiahao Xie, Zhikang Zou, Xiaoqing Ye, Wei Xu, and Xiang Bai. *Crowdclip: Unsupervised crowd counting via vision-language model*. Proceedings of the IEEE/CVF Conference on Computer Vision and Pattern Recognition, pp. 2893–2903, 2023.

---

> ### Author Response · Authors · 2024-11-23
> **Initial response to Reviewer XkFU: part 2**
>
> **Q4. Comparison With State-of-the-Art Methods**
> The comparison methods used in the paper were the latest methods to the best of our knowledge at the time of submission. Since there is no mention of the missing comparisons, we believe the reviewer is referring to "Robust Zero-Shot Crowd Counting and Localization With Adaptive Resolution SAM" [4].
>
> The method uses the foundation model "Segment Everything Everywhere Model" (SEEM) to produce the localization annotations from the segmentation results. Next, these annotations are used to train a crowd-density prediction network to perform both crowd counting and localization. However, [4] is not a fair comparison against the proposed method or annotator-free methods for the following reasons, even though [4] does not rely on ground truth annotations.
>
> 1. SEEM is a promotable panoptic segmentation foundation model. Therefore, SEEM already acts as a weak object detector to produce the localization proposal. Unlike other comparison methods in `table 1`, [4] relies on another downstream task model to produce the annotations.
> 2. SEEM is trained with human interaction in the loop and supervised labels to create semantic awareness. Therefore, [4] is not purely annotation cost-free, unlike the proposed method or other comparisons used in `table 1`.
>
> Therefore, [4] is not a fair comparison against other annotator-free crowd-counting works.
>
> ---
>
> **Q5. Theoretical Foundations**
> To explain the contribution of knowledge augmentation to improving zero-shot crowd counting, we use a probabilistic approach.
>
> The goal is to predict the crowd count $c_i$ for the target embedding $\mathbf{e}_i$. Using a probabilistic framework, the prediction can be expressed as:
>
> $P(c_i|\mathbf{e}_i) = P(c_i|\mathbf{e}_i^{'}),$
>
> where the augmented embedding is:
>
> $\mathbf{e}_i^{'} = \mathbf{e}_i + \mathbf{v}_i^L + \mathbf{v}_i^{LV} + \mathbf{v}_i^{VV}.$
>
> Using Bayes' rule, we can rewrite the probability as follows:
>
> $P(c_i|\mathbf{e}_i^{'}) \propto P(\mathbf{e}_i^{'}|c_i)P(c_i),$
>
> where $P(\mathbf{e}_i^{'}|c_i)$ and $P(c_i)$ denote the likelihood of the augmented embedding given the count and the prior probability of the count derived from the source distribution.
>
> The likelihood can be decomposed as:
>
> $P(\mathbf{e}_i^{'}|c_i) \propto P(\mathbf{e}_i|c_i) \prod_n P(\mathbf{v}_i^n|c_i),$
>
> where $\mathbf{v}_i^n$ is each individual augmentation type from the KAM. However, each individual augmentation is computed from the KAM using the retrieved embeddings from the reference database. Therefore, the likelihood can be updated as:
>
> $P(\mathbf{e}_i^{'}|c_i) \propto P(\mathbf{e}_i|c_i) \prod_n \prod_k^K P(r_k^n|c_i),$
>
> where $r_k^n$ denotes the retrieved embedding augmented with multi-head attention (MHA), and $k$ is the index of the retrieved embedding. Each $\textbf{v}_i^n$ thus encodes the aggregated likelihood information from its corresponding patches, ensuring that $\mathbf{e}_i^{'}$ effectively aligns with the count $c_i$ as MHA behaves as a projection of the query embedding to the key embedding. Consequently, the retrieved embeddings $\textbf{v}_i^n$ encode domain-specific patterns, improving the likelihood estimation.
>
> Without the retrieved embeddings, the likelihood distribution will only depend on $\textbf{e}_i$, and as augmentations are introduced, the likelihood distribution is influenced by the source domain information. The influence of the source likelihood increases with the number of retrieved embeddings. In return, the posterior distribution $P(c_i|\textbf{e}_i^{'})$ becomes a sharper posterior distribution. As the posterior distribution becomes sharper, the uncertainty involved with the prediction reduces, improving the prediction accuracy.
>
>
> ---
>
> **References**
> [4] Jia Wan, Qiangqiang Wu, Wei Lin, and Antoni Chan. *Robust zero-shot crowd counting and localization with adaptive resolution sam*. European Conference on Computer Vision, pp. 478–495. Springer, 2025.

---

> ### Comment · Reviewer_XkFU · 2024-11-25
>
> Thanks for the author's rebuttal.
>
> I have re-read this paper and combined with the comments of other reviewers, but I still deem that this paper does not meet the criteria for acceptance.

---

> > ### Author Response · Authors · 2024-11-25
> >
> > Thank you for reviewing our submission and for providing your feedback. We noticed that your primary concern is the novelty of our work, but we would appreciate more specific details or examples to better understand your perspective. We would be grateful if you could elaborate on specific aspects where you feel our work lacks novelty or overlaps with prior art. Your feedback would greatly help us refine our approach and presentation.
> >
> > Once again, we appreciate your feedback and would welcome any additional insights you may have. We are committed to improving our work and contributing valuable research to the community.

---

> > > ### Author Response · Authors · 2024-11-27
> > > **Revisions to the manuscript**
> > >
> > > **Q2. Detailed Ablation Studies**
> > >     We updated `table 4` to include the baseline performance to compare against the improvements from the KAM. Furthermore, we have provided a more detailed version of the ablation on the effect of different augmentation configurations for the KAM in `table 7` in `section B` of the Appendix.
> > >
> > > **Q3. Computational Complexity Analysis**
> > >     In ' section G ' of the Appendix, we have provided the computational complexity associated with the proposed method compared to the baseline architecture. Additionally, we provide the computational efficiency analysis in `section F` for the retrieval process as per reviewer oxEJ's comments to understand the practical implications of the proposed method.
> > >
> > > **Q5. Theoretical Foundations**
> > >     In `section H` of the Appendix, we have provided a theoretical explanation for performance improvement using knowledge augmentation with a probabilistic approach.

---

### Official Review · Reviewer_VY3u · 2024-10-31

**Soundness:** 2
**Presentation:** 1
**Contribution:** 2
**Rating:** 3
**Confidence:** 3

**Summary:**

This paper presents a zero-shot counting method based on synthetic data and vision-language models.

**Strengths:**

The authors claim that this study is the first one to combine synthetic data and VLM, achieving better performance than other LVM counting models.

**Weaknesses:**

Most of the paper is difficult to understand. Many variables are not well-defined, and the corresponding descriptions are confusing.

**Questions:**

- What is the maximum inner product? What is its formulation?
- Section 3.2 is hard to understand. What is retrieved in this part? If the retrieval process is considered a black box, what is the input for the retrieval? What is the output of the retrieval? What is the database used for the retrieval?
- On line 259, how is the image encoder \(\Psi\) and text encoder \(\Phi\) pretrained? Are they derived from CLIP? If so, please add relevant descriptions and citations.
- Where does the image embedding \(\mathbf{e}_i\) come from?
- The pipeline is unclear. A figure demonstrating the overall process is required.
- What is the post-processing of \(\mathbf{e}'_i\) in Eq. (6)?
- In Eq. (7), how is \(\hat{c}_i\) estimated, and what is its ground truth (GT)? There is no prior reference or description explaining their source.

---

> ### Comment · Reviewer_VY3u · 2024-11-14
> **read again**
>
> I reread the paper, and now I can understand most of the problems I commented on above. However, the writing should be improved.
>
> The following problems still confused me:
>
> 1. The text description contains three keys: [weather condition], [crowd count], and [time of day]. Is there any ablation study demonstrating how these three keys affect the retrieval effectiveness?
> 2. How is KAM supervised? the output of KAM is a embedding (\mathbf{e}_i) in Eq.(6), there is not connection between it and the count c_i in Eq.(7).
> 3. In Table 4, the ablation studies on (v_i^{L}), (v_i^{LV}), and (v_i^{VV}) are conducted. If all of them are not involved, but only (\mathbf{e}_i) is used as the embedding, how is the performance?

---

> ### Author Response · Authors · 2024-11-23
> **Initial response to Reviewer VY3u**
>
> Thank you for your insightful and detailed feedback on our manuscript. We appreciate your suggestions to include an ablation study on the impact of the three text keys, clarify the supervision mechanism for KAM, and analyze the performance when only the embedding $(\mathbf{e}_i)$ is used.
>
> **Q1. Ablation Study on Keywords**
> The retrieval process only considers the image embeddings to perform the maximum inner-product search and extract the closest neighbors. The text description is not used in the retrieval process and is only used in the knowledge augmentation step. Also, we did not perform an ablation on the effect of the three keywords in our experiments originally. However, we agree that ablation of the effect of the keywords is beneficial to understand whether the additional context is important, as suggested by the reviewer.
>
> We performed three experiments with the Knowledge Augmentation Module (KAM) setting corresponding to row 4 in `table 4`. The first ablation was performed with only the "crowd count" as the keyword. Then, we considered the "time of day" as the only additional keyword. Lastly, we considered the "weather" as the only additional keyword.
>
> **Table A**: Ablation of keywords.
>
> | Method            | JHU   | SHA   | QNRF  |
> |-------------------|-------|-------|-------|
> | baseline          | 142.3 | 118.4 | 214.9 |
> | count             | 147.5 | 117.6 | 215.8 |
> | + time of day     | 143.5 | 118.1 | 215.1 |
> | + weather         | 147.8 | 118.3 | 216.1 |
>
> The performance for the three ablations was similar to the baseline performance for the ShanghaiTech-A (SHA) and UCF-QNRF (QNRF) datasets. Therefore, the additional context keywords did not significantly affect the performance. However, for the JHU-Crowd++ (JHU) dataset, using the "time of day" keyword improved the performance, unlike the "weather" keyword. Therefore, the "weather" keyword could be removed from the text prompt even though there is no significant improvement to the inference speed or computational cost. However, the "time of day" keyword cannot be simply discarded.
>
> The difference between SHA, QNRF, and JHU is that the JHU is a large dataset compared to the other two and contains a distribution of scene illuminations, whereas the other two datasets mostly have bright scenes. Since the training of the KAM and the count decoder are similar among these three datasets, the performance difference should come from the ability of the model to generalize. Since SHA and QNRF have a skewed illumination distribution, the "time of day" keyword has no effect. But, for the JHU dataset, the "time of day" keyword can provide context on the level of illumination as the "morning" and "evening" labels allude to bright illuminations, as opposed to the "night" label.
>
> We will add the ablation of the effect of the keywords on performance in the revised version.
>
> ---
>
> **Q2. Supervision Mechanism for KAM**
> The KAM is supervised during the fine-tuning stage along with the count decoder. The embedding $\mathbf{e}_i$ is the output of the image encoder, and the final output of the KAM is the augmented image embedding $\mathbf{e}_i^{'}$ given by `equation (6)`. The input to the count decoder is $\mathbf{e}_i^{'}$, which connects the KAM and count $c_i$ to supervise the augmentation module. Hence, the relationship between $c_i$ and $\textbf{e}_i$ can be written as $c_i = f(\mathbf{e}_i + \mathbf{v}_i^L + \mathbf{v}_i^{LV} + \mathbf{v}_i^{VV}),$
> where $f$ is the `count decoder`.
>
> ---
>
> **Q3. Performance with $\mathbf{e}_i$ Only**
> When the augmentations are removed, the input to the count decoder will be the output of the image encoder, which is $\mathbf{e}_i$, and the count decoder is then fine-tuned to map $\mathbf{e}_i$ to $c_i$ using the simulated or synthetic data. This is the baseline method to analyze the contribution of the KAM and different augmentation methods. The performance of the baseline method is given in row 4 of `table 3`. However, we will add the performance of the baseline method to `table 4` to improve the informativeness of the table in the revised version.

---

> > ### Comment · Reviewer_VY3u · 2024-11-26
> >
> > The following questions are not addressed:
> >
> > - In lin 238, what is the maximum inner product? What is its formulation?
> > - On line 259, how is the image encoder $\Psi$ and text encoder $\Phi$ trained? Are they derived from CLIP? If so, please add relevant descriptions and citations.
> > - In line 268, Where does the image embedding $\mathbf{e}_i$ come from?
> > - The pipeline is unclear. A figure demonstrating the overall process is required.
> > - What is the post-processing of $\mathbf{e}'_i$ in Eq. (6)?
> > -  In Eq. (7), how is (\hat{c}_i) estimated, and what is its ground truth (GT)? There is no prior reference or description explaining their source.
> > ---
> > Some parts should be addressed to improve reading:
> >
> > 1. A figure should be present rather than only a description for the two-stage retrieval (real and then synthetic).

---

> > > ### Author Response · Authors · 2024-11-27
> > > **Second response to Reviewer VY3u: part 1**
> > >
> > > We used $c_i$ to denote the estimated count in our previous response. Below are the corrected responses.
> > >
> > > **Q2. Supervision Mechanism for KAM**
> > > The KAM is supervised during the fine-tuning stage along with the count decoder. The embedding $\mathbf{e}_i$ is the output of the image encoder, and the final output of the KAM is the augmented image embedding $\mathbf{e}_i^{'}$ given by `equation (6)`. The input to the count decoder is $\mathbf{e}_i^{'}$, which connects the KAM and count $\hat{c}_i$ to supervise the augmentation module. Hence, the relationship between $\hat{c}_i$ and $\textbf{e}_i$ can be written as $\hat{c}_i = f(\mathbf{e}_i + \mathbf{v}_i^L + \mathbf{v}_i^{LV} + \mathbf{v}_i^{VV}),$
> > > where $f$ is the `count decoder`. Then, during the training stage of the KAM and the count decoder, we have the ground truth counts of $c_i$ for the simulated images. Hence, we can compute the loss between the prediction and the ground truth as mentioned in `equation 7`.
> > >
> > > ---
> > >
> > > **Q3. Performance with $\mathbf{e}_i$ Only**
> > > When the augmentations are removed, the input to the count decoder will be the output of the image encoder, which is $\mathbf{e}_i$, and the count decoder is then fine-tuned to map $\mathbf{e}_i$ to $\hat{c}_i$ using the simulated or synthetic data. This is the baseline method to analyze the contribution of the KAM and different augmentation methods. The performance of the baseline method is given in row 4 of `table 3`. However, we will add the performance of the baseline method to `table 4` to improve the informativeness of the table in the revised version.
> > >
> > > ---
> > >
> > > Next, we provide our responses to the reviewer's remaining questions.
> > >
> > > **Q4. Maximum inner product search**
> > > Maximum Inner Product Search (MIPS) is a well-known problem formulation used primarily in information retrieval, machine learning, and recommendation systems [1]. The goal is to find the vector in a dataset that has the maximum inner product with a given query vector.
> > >
> > > Let:
> > >
> > > **Dataset**:
> > >
> > >         \[\mathcal{X} = \{ \mathbf{x}_1, \mathbf{x}_2, \ldots, \mathbf{x}_N \} \subseteq \mathbb{R}^d,\]
> > > where $\textbf{x}_i$ are the data vectors.
> > >
> > > **Query**:
> > >
> > >         \[\mathbf{q} \in \mathbb{R}^d,\]
> > > the query vector.
> > >
> > > **Inner Product**:
> > >
> > >         \[\langle \mathbf{q}, \mathbf{x} \rangle = \sum_{j=1}^d q_j x_j,\]
> > > the dot product between the query and a dataset vector.
> > >
> > > The objective is to find the dataset vector $\mathbf{x}^* \in \mathcal{X}$ that maximizes the inner product with $\mathbf{q}$:
> > >
> > >         \[\mathbf{x}^* = \arg\max_{\mathbf{x} \in \mathcal{X}} \langle \mathbf{q}, \mathbf{x} \rangle.\]
> > >
> > > ---
> > >
> > > **Q5. Image and text encoder training**
> > > The image encoder $\Psi$ and the text encoder $\Phi$ are pre-trained models. The architectures used for $\Psi$ and $\Phi$ are provided in section 4.1 with the appropriate citations. $\Psi$ is ViT-B/16 and $\Phi$ SentenceT model as mentioned in `section 4.1`. Both $\Psi$ and $\Phi$ are pre-trained and not fine-tuned during the training of the KAM and the count decoder, and this has been mentioned in `lines 252-253`  of the revised version. $\Psi$ is pre-trained, as mentioned in `section 3.1`.
> > >
> > > ---
> > >
> > > **Q6. Producing image embedding $\textbf{e}_i$**
> > > The image embedding $\textbf{e}_i$ is produced by passing the input image $I_i$ through the pre-trained image encoder $\Psi$. We have clarified this in `lines 225-226` in the revised version. Further, we illustrate the generation of $\textbf{e}_i$ in the overall pipeline figure in `figure 2` of the revised version.
> > >
> > > ---
> > >
> > > **Q7. Overall pipeline illustration**
> > > We have illustrated the overall pipeline in `figure 2` of the revised version.
> > >
> > > ---
> > >
> > > **References**
> > > [1] Hsiang-Fu Yu, Cho-Jui Hsieh, Qi Lei, and Inderjit S Dhillon. *A greedy approach for budgeted maximum inner product search*. Advances in neural information processing systems, 30, 2017.

---

> > > > ### Author Response · Authors · 2024-11-27
> > > > **Second response to Reviewer VY3u: part 2**
> > > >
> > > > **Q8. Post-processing in Eq. 6**
> > > > In `equation 6` to produce augmented image embeddings $\textbf{e}_i^{'}$ from the input image embeddings $\textbf{e}_i$. The remaining variables are the three different augmentation features extracted from the Knowledge Augmentation Module (KAM). `Equation 6` adds the features extracted from the three augmentations to $\textbf{e}_i$. The post-processing of $\textbf{e}_i^{'}$ is to feed the feature vector to the count decoder to produce the estimated count $\hat{c}_i$.
> > > >
> > > > ---
> > > >
> > > > **Q9. Clarification of $c_i$ and $\hat{c}_i$**
> > > > The ground truth and estimated count in `equation 7` are denoted by the variables $c_i$ and $\hat{c}_i$ as stated in `lines 290-291`. The ground truth counts are used to train the KAM and the count decoder. In this stage, we use the simulated data as the training dataset for which we have the labeled data. Hence, we have the ground truth data $c_i$ for the simulated data, and $\hat{c}_i$ is the output of the count decoder.
> > > >
> > > > ---
> > > >
> > > > **Q10. Two-stage retrieval process**
> > > > The two-stage retrieval process is illustrated in the top block of `figure 3a` and we describe it under the figure caption. However, we include a more illustrative figure of the two-stage retrieval process in the Appendix.

---

> > > > > ### Author Response · Authors · 2024-11-27
> > > > > **Revisions to the manuscript**
> > > > >
> > > > > **Q1. Ablation Study on Keywords**
> > > > >     We included a new paragraph, **Effect of the text description**, in `lines 513 — 520` to study the performance of the keywords used in the text description. We also tabulated these results in `table 5` of the revised version.
> > > > >
> > > > > **Q2. Supervision Mechanism for KAM**
> > > > >     We added the `line 275` to mention how $\textbf{e}_i^{'}$ and $\hat{c}_i$ are connected to supervise the KAM and the count decoder. Furthermore, $c_i$ and $\hat{c}_i$ are the ground truth and estimated counts for the simulated images. This was already explained in `lines 301 - 302` of the original version and is in lines 290 - 291 of the revised version.
> > > > >
> > > > > **Q3. Performance with $\mathbf{e}_i$ Only**
> > > > >     We updated `table 4` to include the baseline performance (i.e. without any knowledge augmentations).
> > > > >
> > > > > **Q4. Maximum inner product search**
> > > > >     Maximum inner product search (MIPS) is a well-known problem formulation. Hence, we have added the citation referring to MIPS in `line 227` of the revised version.
> > > > >
> > > > > **Q6. Producing image embedding $\textbf{e}_i$**
> > > > >     We have clarified this in `lines 225-226` in the revised version. Further, we illustrate the generation of $\textbf{e}_i$ in the overall pipeline figure in `figure 2` of the revised version.
> > > > >
> > > > > **Q7. Overall pipeline illustration**
> > > > >     We have illustrated the overall pipeline in `figure 2` of the revised version.
> > > > >
> > > > > **Q8. Post-processing in Eq. 6**
> > > > >     We have added the `line 275` with regard to the post-processing $\textbf{e}_i^{'}$ in `equation 6` in the revised version.
> > > > >
> > > > > **Q10. Two-stage retrieval process**
> > > > >     We added an illustrative description of the two-stage retrieval process in `figure 7` in `section A` of the Appendix.

---

> ### Author Response · Authors · 2024-11-25
>
> We sincerely believe we have addressed the reviewer’s questions, including thorough explanations, additional ablations, and justifications.
>
> As the deadline for the discussion period approaches (November 26th), we wanted to gently follow up to see if you've had the opportunity to review our response.
>
> If you have any further questions or need clarification, please let us know. We would be more than happy to provide detailed answers to ensure all concerns are fully addressed.

---

> ### Comment · Reviewer_VY3u · 2024-11-27
>
> **FQ2 & Q9:** Does the ground truth, $ c_i $, come from the retrieved synthetic label? There are $ K $ synthetic image-text pairs acquired. How is $ c_i $ computed from these $ K $ pairs? I did not find any description of this in the paper.
>
> **FQ4:** You should not assume that your readers know how the Maximum Inner Product (MIP) is computed. In Sec. 3.2, you should describe the formulation of it, rather than just replying to me.
>
> **FQ7:** The pipeline presented in Fig. 2 greatly reduces the difficulty in understanding the paper. Nice figure.
>
> **FQ10:** Fig. 7 is too abstract. It would be better to use synthetic or real crowd samples to illustrate the process.
>
> **Q11:** This model requires too many transformer modules. Table 9 presents the efficiency, but it should be compared with other models.
>
> **Q12:** The pipeline is elaborate, but the performance is not as good as some previous models, such as DANN (in 2019) and BLA (in 2022). Using the same training data (the encoder is also trained with extra data from other real datasets), what is the advantage of the proposed method?
>
> - DANN: "Domain-adaptive Crowd Counting via High-quality Image Translation and Density Reconstruction," Junyu Gao, et al., 2019.
> - BLA: "Bi-level Alignment for Cross-Domain Crowd Counting," Shenjian Gong, et al., 2022.

---

> ### Author Response · Authors · 2024-11-27
> **Third response to Reviewer VY3u and revision changes**
>
> **FQ2 & Q9. Ground truth source**
> For simulated data, we have the ground truth data unlike for real data. For each image in the simulated dataset, we know the crowd count, weather conditions, and time of day. As mentioned in `lines 191 - 192`, 80$\%$ of the simulated data will be used for the reference dataset as well as pre-train the image encoder, and the remaining 20$\%$ is used to train the KAM and count decoder. As aforementioned, we have the ground truths for all the images in the simulated data. Hence, we know the crowd count of each image in the 20$\%$ training set. Therefore, we don't need to estimate $c_i$ from the retrieved pairs for training images as we readily have the count.
>
> - We added `lines 289 - 292` to mention the source of $c_i$ required for the loss computation in `equation 7`.
>
> ---
>
> **FQ4. MIPS formulation**
>
> - We have added the `lines 220 - 225` to formulate the MIPS.
>
> ---
>
> **FQ7. Pipeline figure**
>
> We appreciate your feedback to improve the presentation of the pipeline and the paper.
>
> ---
>
> **FQ10. Two-stage retrieval process**
> - We have added `figure 8` in the appendix with real and simulated crowd samples to illustrate the process. Additionally, we included `lines 752 - 755` elaborating `figure 8` for the two-stage retrieval process.
>
> ---
>
> **Q11. Computational efficiency**
> In `table 9` GFLOPS measures the rate at which a computing system can execute floating-point operations. This rate is influenced by the number of retrieved data we feed into the KAM module. Which is why the GLOPS is higher than the baseline. However, for a single retrieved image-text pair, the GLOPS is 4.368. Besides that, we compare the inference speed with other models, such as CSS-CCNN and CrowdCLIP, in `table 8` which is critical for practical applications. As a fellow transformer-based model, crowd clip has a higher number of transformer modules compared to our architecture since CrowdCLIP uses two ViT-B/16 for visual encoding in addition to the transformer-based text encoder.
>
> - We have added this description in `lines 915 - 928` in `section G` of the Appendix.
>
> ---
>
> **Q12. Comparison with domain adaptation**
> We agree that both methods use the GCC dataset and real data to train. However, both methods are domain-adaptation crowd-counting methods, whereas our method is domain-general crowd-counting. Domain general crowd-counting aims to estimate the crowd count for unseen image distributions. In domain adaptation, a model is trained on one dataset (source) to perform well on a target dataset. Domain adaptation tries to bridge the performance gap caused by the domain shift. In both DANN and BLA, the network is trained for each real dataset and tested for the particular dataset. Hence, in both these methods, the model already has seen the target distribution. However, in our method, we do not use the real dataset that we are testing in any way. Hence, the testing distribution is truly unknown. Therefore, it is unfair to compare the performance of domain adaptive methods against domain generalized methods.
>
> For a fair comparison, these two methods should be evaluated for cross-dataset performance. For example, the model will be trained with GCC and SHA datasets and will be tested on the SHB dataset. Then, we can compare the performance with our proposed method. Neither method has released the training code or models for the datasets used. Regardless, if we compare the performance drop in our proposed method compared to BLA (beats DANN in SHA, SHB, and QNRF), the percentage-wise reduction is $16$%, $50$%, and $7$% for SHA, SHB, and QNRF, respectively. Out of the three datasets, QNRF is the most diverse compared to SHB and SHA. BLA was only able to get a $7$% improvement by having access to the training for QNRF, whereas the proposed method does not require the training set of the test distribution. Besides that, we compare the performance of existing methods under the domain-general setting in `table 3` using the GCC dataset.

---

### Official Review · Reviewer_oxEJ · 2024-11-01

**Soundness:** 2
**Presentation:** 2
**Contribution:** 2
**Rating:** 5
**Confidence:** 3

**Summary:**

This paper proposes ReZeS-Count, a retrieval-based zero-shot crowd counting approach. It leverages external knowledge retrieval, extracting image-text pairs from a simulated crowd dataset to augment real crowd image embeddings. Experiments demonstrate that the ReZeS-Count outperforms current zero-shot crowd counting approaches.

**Strengths:**

1. External information is relatively easy to obtain compared to detail annotations. This method makes the first attempt to utilize external information during testing for improving crowd counting accuracy, reducing reliance on labeled data.
2. Experiments illustrate that the proposed zero-shot method achieves competitive performance compared with the fully-supervised method MCNN.

**Weaknesses:**

1. Lack of efficiency analysis in the manuscript. Will this approach become time-consuming as the retrieval space scales?
2. What is the baseline method? As shown in Table 4, even the model in line 1 surpasses the fully supervised method MCNN. Which design component is the most critical for achieving this?
3. The presentation of this manuscript could be improved. What is the structure of the count decoder? According to Figure 3, the decoded values are integers. It is not clear how to decode these integer counts.

**Questions:**

Please refer to the weaknesses.

---

> ### Author Response · Authors · 2024-11-23
> **Initial response to Reviewer oxEJ**
>
> Thank you for your valuable feedback on our manuscript. We appreciate your suggestions for conducting an efficiency analysis, clarifying the baseline method and its critical design components, and improving the explanation of the count decoder structure.
>
> **Q1. Efficiency Analysis**
> For the retrieval process, we use the naive maximum inner product search. This involves computing the similarity between image embeddings and crop embeddings in the reference database and sorting the similarity scores to find the closest neighbors.
>
> Suppose the reference database is of size $N$, the embedding dimensionality is of size $d$, and we need to find the nearest $k$ neighbors. Then, the computational efficacy of the whole process is $O(N \cdot d \ + \ N \cdot \log k)$. Accordingly, as the retrieval space scales, the time it takes for the retrieval process will increase. However, for larger reference databases, using approximation methods such as the k-d tree, the computational complexity can be reduced to $O(\log N)$ for smaller dimensional sizes, but still, the time consumed will increase with the size of the reference database.
>
> ---
>
> **Q2. Baseline Method and Augmentations**
> The crucial module of the pipeline is the Knowledge Augmentation Module (KAM). In the KAM, we use three different embedding augmentations, as depicted by the first three columns of `table 4`. Therefore, the baseline method to compare the performance of different augmentation types would be the counting performance without any kind of augmentation.
>
> For the baseline method, the image encoder is pre-trained with both simulated and real data, as described in `section 3.1`. Then, the count decoder is trained with the simulated data without any image-text retrieval and knowledge enhancement. The performance of the baseline method is given in the fourth row of `table 3`. We will update `table 4` to include the baseline performance.
>
> The proposed architecture's baseline method surpasses the performance of the fully supervised method MCNN. MCNN uses multi-scale convolutional networks with three different receptive fields and maps the image to a downscaled density map. MCNN is a CNN-based network where the receptive field is mostly local, in contrast to the receptive field of a transformer-based network. Furthermore, it should be noted that CrowdCLIP performs closer to MCNN even though CrowdCLIP has an inherent error introduced by the discrete counting levels. This shows the primitive nature of methods like MCNN, where the latent space is directly compared to the density space without any projection.
>
> The performance gain by each augmentation type is provided for only $\mathbf{v}_i^{L}$ in `table 4`. Therefore, to understand which augmentation types improve the performance, we provide the counting performance for each augmentation type compared against the baseline performance in the following table.
>
> **Table A**: Performance ablation for KAM.
> | Augmentation       | JHU     | SHA     | QNRF    |
> |--------------------|---------|---------|---------|
> | MCNN               | 188.9   | 110.2   | 277.0   |
> | baseline           | 170.2   | 143.1   | 223.6   |
> | $\mathbf{v}_i^{L}$  | 148.6  | 123.6   | 215.4   |
> | $\mathbf{v}_i^{LV}$ | 152.8  | 128.3   | 219.8   |
> | $\mathbf{v}_i^{VV}$ | 145.3  | 125.8   | 216.5   |
>
> The most performance gain has come from the components $\mathbf{v}_i^{L}$ and $\mathbf{v}_i^{VV}$ compared to the baseline method. The two augmentations deliver text information and visual information, respectively, but the cross-attention is taken between the image embeddings and retrieved patch embeddings. However, the performance gain from $\mathbf{v}_i^{LV}$ is less compared to the other two augmentations where the cross-attention is taken between the image and retrieved text embeddings.
>
> ---
>
> **Q3. Count Decoder Structure**
> For this two-part question, we will first address the structure of the count decoder. In our proposed work, we use a simple `MLP` with a single `Linear` layer following Transcrowd [1]. Secondly, the decoded values of the count decoder are real numbers and not explicitly integers. However, we rounded the decoded values to the closest integer in `figure 3`, and there is no constraint imposed on the decoder to produce integer values for the estimated count.
>
> ---
>
> **References**
> [1] Dingkang Liang, Xiwu Chen, Wei Xu, Yu Zhou, and Xiang Bai. Transcrowd: Weakly-supervised crowd counting with transformers. *Science China Information Sciences*, 65(6):1–14, 2022.

---

> ### Author Response · Authors · 2024-11-25
>
> We sincerely believe we have addressed the reviewer’s questions, including thorough explanations, additional ablations, and justifications.
>
> As the deadline for the discussion period approaches (November 26th), we wanted to gently follow up to see if you've had the opportunity to review our response.
>
> If you have any further questions or need clarification, please let us know. We would be more than happy to provide detailed answers to ensure all concerns are fully addressed.

---

> ### Author Response · Authors · 2024-11-27
> **Revisions to the manuscript**
>
> **Q1. Efficiency Analysis**
>     We have provided the computational efficiency analysis for the retrieval process. However, due to space constraints, we have included the analysis in `section F` of the Appendix. Furthermore, we have included the computational cost associated with the proposed method compared to the baseline architecture in `section G` of the Appendix as per the comments of reviewer XkFU.
>
> **Q2. Baseline Method and Augmentations**
>     We updated `table 4` to include the baseline method. Furthermore, we have provided a more detailed version of the ablation on the effect of different augmentation configurations for the KAM in `table 7` in `section B` of the Appendix.
>
> **Q3. Count Decoder Structure**
>     We updated `lines 303 - 304` to indicate the structure of the count decoder.

---

### Official Review · Reviewer_anXR · 2024-11-03

**Soundness:** 3
**Presentation:** 3
**Contribution:** 3
**Rating:** 6
**Confidence:** 5

**Summary:**

This paper proposes ReZeS-Count, a retrieval-based framework for crowd counting under zero-shot conditions by leveraging both visual and textual information from a simulated dataset. The approach employs knowledge retrieval to incorporate multi-modal data, enhancing the inference capabilities of the model on real, unlabeled crowd images. Extensive experiments demonstrate that ReZeS-Count achieves state-of-the-art performance across multiple datasets, showcasing the effectiveness of retrieval-based zero-shot learning for crowd counting.

**Strengths:**

1. This approach effectively bridges the gap between simulated and real data by leveraging multi-modal retrieval and weakly-supervised learning.

2. ReZeS-Count outperforms existing annotator-free and self-supervised methods across multiple public datasets, establishing itself as a robust zero-shot crowd-counting method.

3. The paper provides extensive ablation studies, assessing the impact of various components and retrieval configurations, which offers valuable insights into the effectiveness of different aspects of the model.

**Weaknesses:**

1. The title “Retrieval-Based Zero-Shot Crowd Counting” may not fully align with the actual methodology presented. The approach is more akin to cross-dataset crowd counting rather than true zero-shot learning, as the model still relies on labeled data from a different (simulated) domain. This creates a somewhat misleading impression of the generalization capabilities claimed in the paper. A revised title that reflects the cross-dataset nature of the problem might be more accurate.

2. The paper’s current comparison tables could lead readers to assume that ReZeS-Count is a purely unsupervised method, akin to CrowdCLIP and similar approaches, while it actually uses quantity annotations from the simulated dataset. This reliance on labeled synthetic data should be clarified explicitly. Adding a new column to the tables that indicates the type of annotation (e.g., labeled synthetic data vs. unlabeled real data) would help clarify the supervision levels used by each method. This additional detail is crucial for fair comparisons and to avoid misinterpretation of the method’s level of supervision relative to other “annotator-free” approaches.

3. The paper would benefit from a more thorough qualitative analysis of the features learned by ReZeS-Count. Specifically, an investigation into which visual and textual features are most influential in the retrieval process could enhance understanding of the model’s zero-shot capabilities.

**Questions:**

See the weakness.

If the authors address the above concerns, I would support the acceptance of this paper.

---

> ### Author Response · Authors · 2024-11-23
> **Initial response to Reviewer anXR**
>
> We appreciate the constructive feedback and recognize the proposed method's strengths. Your suggestions for improving the title, content, and qualitative analysis will enhance the clarity of our work.
>
> **Q1. Title Revision**:
>    We agree with the fact that the approach is similar to cross-dataset crowd-counting, and it uses the paired data from the simulated domain. However, we used "zero-shot" in the paper title as the proposed method is not privy to the training data corresponding to the test dataset. Nevertheless, we agree that changing the title to convey the generalization capabilities of the proposed method will be helpful for the reader. Following the paper titled *Domain-General Crowd Counting in Unseen Scenarios* [1], where inference is performed without seeing the training data of the test dataset, we believe an appropriate title for the proposed work would be **Retrieval-based Generalized Crowd Counting**.
>
> ---
>
> **Q2. Tabulation Improvements**:
>    We appreciate the reviewer's suggestion to improve the tables' informativeness to the reader. In the revised version, we will modify the tables to indicate the type of data used in each method.
>
> ---
>
> **Q3. Qualitative Analysis**:
>    We acknowledge that a qualitative analysis could enhance the understanding of the inner workings of the proposed pipeline. However, instead of the features learned, we think that the behavior of the cross-attention maps would be more suitable for understanding the most influential aspects of the pipeline, especially in the Knowledge Augmentation Module.
>
>    For example, if we consider the attention weights when computing **vᵢˡᵛ**, we take the cross-attention between the image embeddings of the test image and the text embeddings of the retrieved description. By averaging and normalizing the attention weights, we could compute attention scores assigned to each word token.
>
>    For instance, we considered an image crop of an indoor scene with low illumination. A retrieved text description for the test crop was:
>
>    > *The image has a clear weather with 52 people in the night.*
>
>    For this example, the attention scores (rounded to the nearest third decimal) produced for each word token were:
>
>    - **The**: `0.000`
>    - **image**: `0.002`
>    - **has**: `0.000`
>    - **a**: `0.000`
>    - **clear**: `0.030`
>    - **weather**: `0.000`
>    - **with**: `0.000`
>    - **52**: `0.905`
>    - **people**: `0.003`
>    - **in**: `0.000`
>    - **the**: `0.000`
>    - **night**: `0.060`
>
>    The attention scores are high for the crowd count, time of day, and weather, as these three keywords carry information among different images because the remaining text words are common across all text descriptions. Since both test image and retrieval embeddings are generated from the same encoder, these attention scores highlight which feature maps are more influential due to the way the attention mechanism is developed.
>
>    Furthermore, we will include cross-attention maps for the retrieved text description in the revised draft.
>
> ---
>
> **References**:
> [1] Zhipeng Du, Jiankang Deng, and Miaojing Shi. Domain-general crowd counting in unseen scenarios. *Proceedings of the AAAI Conference on Artificial Intelligence*, volume 37, pp. 561–570, 2023.

---

> ### Author Response · Authors · 2024-11-25
>
> We sincerely believe we have addressed the reviewer’s questions, including thorough explanations, additional ablations, and justifications.
>
> As the discussion period deadline approaches (November 26th), we wanted to follow up and see if you've had the opportunity to review our response.
>
> If you have any further questions or need clarification, please let us know. We would be more than happy to provide detailed answers to ensure all concerns are fully addressed.

---

> > ### Comment · Reviewer_anXR · 2024-11-26
> >
> > Sorry, I do not see the revised version

---

> ### Author Response · Authors · 2024-11-27
> **Revisions to the manuscript**
>
> **Q1. Title Revision**:
>    We have changed the title of the manuscript to **Retrieval-based Generalized Crowd Counting**.
>
> **Q2. Tabulation Improvements**:
>    We have modified `Table 1` and `Table 2` to accommodate your suggestions.
>
> **Q3. Qualitative Analysis**:
>    We included a new paragraph, **Qualitative analysis**, in `lines 447 - 465` to enhance the understanding of the proposed method's functioning with a qualitative approach. Further, we provide `figure 6` for the qualitative analysis.

---

### Meta-Review · Area_Chair_wxzi · 2024-12-19

**Metareview:**

This paper was reviewed by four experts in the field. The ratings are 5,3,6,5. This paper proposes a retrievel-based approach for crowd counting that uses both synthetic and real image datasets.

Overall, reviewers agree that this paper proposes an interesting approach leveraging synthetic data and VLM for crowd-counting without full supervision. However, they also expressed many concerns, e.g. clarity, ablation, computation complexity, etc. The rebuttal addressed some of the concerns, but it did not successfully assuage reviewers' opinions. There is no ground to overrule reviewers' recommendations.

**Additional Comments On Reviewer Discussion:**

The author rebuttal addressed some of the concerns of the reviewers. But it does not successfully assuage any reviewers' opinions.

---

### Decision · Program_Chairs · 2025-01-22

Reject